# Controlling Thinking Speed in Reasoning Models

**Zhengkai Lin**[1], **Zhihang Fu**[2]*, **Ze Chen**[2], **Chao Chen**[2], **Liang Xie**[3],
**Wenxiao Wang**[4]*, **Deng Cai**[1], **Zheng Wang**[2], **Jieping Ye**[2]

[1]State Key Lab of CAD&CG, Zhejiang University,   [2]Alibaba Cloud
[3]College of Computer Science and Technology, Zhejiang University of Technology
[4]School of Software Technology, Zhejiang University

## Abstract

Human cognition is theorized to operate in two modes: fast, intuitive System 1 thinking and slow, deliberate System 2 thinking. While current Large Reasoning Models (LRMs) excel at System 2 thinking, their inability to perform fast thinking leads to high computational overhead and latency. In this work, we enable LRMs to approximate human intelligence through *dynamic thinking speed adjustment, optimizing accuracy-efficiency trade-offs*. Our approach addresses two key questions: (1) how to control thinking speed in LRMs, and (2) when to adjust it for optimal performance. For the first question, we identify the steering vector that governs slow-fast thinking transitions in LRMs' representation space. Using this vector, we achieve the first representation editing-based test-time scaling effect, outperforming existing prompt-based scaling methods. For the second question, we apply real-time difficulty estimation to signal reasoning segments of varying complexity. Combining these techniques, we propose the first reasoning strategy that enables fast processing of easy steps and deeper analysis for complex reasoning. Without any training or additional cost, our plug-in module delivers an average +1.3% accuracy with -8.6% token usage across leading LRMs and advanced reasoning benchmarks. All of our algorithms are implemented based on vLLM and are expected to support broader applications and inspire future research.[1]

## 1  Introduction

Cognitive theory categorizes human thinking into two systems: the fast, intuitive System 1 and the slower, deliberate System 2 [39, 13]. Large Language Models (LLMs) exhibit similar specialization. The chain-of-thought (CoT) [26, 40] generated by models like GPT-4o [10] and DeepSeek-V3 [6] exhibits fast, intuitive thinking that is ideal for routine tasks but limited in advanced reasoning. In contrast, the slower, more deliberate long thinking utilized by Large Reasoning Models (LRMs), such as DeepSeek-R1 [7] and OpenAI's o1 series [11], demonstrates superior performance on advanced tasks. However, this advantage comes at the cost of largely increased computational overhead. This contrast motivates our core research question: *How can we combine the advantages of both System 1 and System 2 thinking within one model, thus simultaneously achieving both efficiency and accuracy?*

Prior work integrates System 1 and System 2 thinking into one model by fine-tuning on both fast and slow reasoning traces [28, 33]. Alternatively, we argue that even **without training**, some LRMs **intrinsically possess** both slow- and fast-thinking abilities. Through statistical analysis of LRMs' responses, we notice that the slow and fast outputs consistently start with distinct opening words. This built-in feature provides a natural switch for activating different thinking modes within a single LRM. However, *achieving human-level cognition* requires not only the possession of both System 1 and System 2 thinking modes, but also the capacity for *dynamic transitions* between the 2 modes during thinking. Building on these insights, we refine our core research focus into two key issues:

---

*Corresponding authors. Email: zhihang.fzh@alibaba-inc.com, wenxiaowang@zju.edu.cn.

[1]Code available at: `https://github.com/D2I-ai/thinking-speed-control`

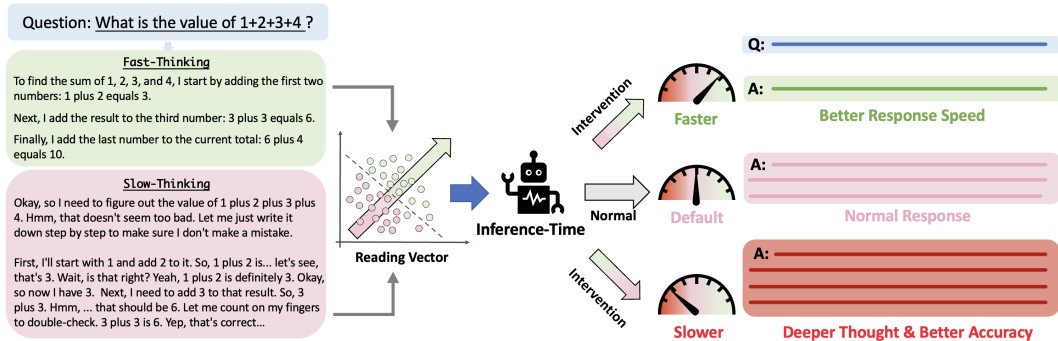

Figure 1: The framework of our thinking speed control method. We utilize the System 1 and System 2 thinking modes of LRMs to sample fast- and slow-thought pairs. We then extract the steering vector that governs the transition between these thinking modes from LRMs, which enables us to steer the LRMs toward either **fast-thinking** for efficiency or **slow-thinking** for better accuracy.

1. *How can we implement transitions between slow- and fast-thinking during LRMs' reasoning?*
2. *When should we transition to fully utilize the advantages of both thinking modes?*

To address the first issue, we identify LRMs' internal representations for different thinking modes using representation engineering techniques [45]. As shown in Figure 1, the extracted reading vector enables smooth control over the thinking modes, steering LRMs towards either **faster thinking** (producing concise responses) or **slower thinking** (generating complex outputs). We term this method **"thinking speed control"**, as it governs LRMs' thinking behaviors (*i.e.*, the succinctness of each reasoning step) and thus determines the efficiency of their derivations of final answers. We demonstrate the effectiveness of our method through a promising inference-time **scaling effect**: (1) responses become **more concise** under fast-thinking steering while maintaining superior performance compared to traditional budget-based methods, and (2) the thoughts become **longer** and accuracy gradually improves under slow-thinking steering, successfully surpassing both the original performances of LRMs and other inference-time scaling methods.

For the second issue, ideally, one would expect LRMs to quickly skim over the easy parts of the reasoning and spend more time (tokens) on conquering the difficult parts. Guided by this intuition, we first propose a real-time reasoning difficulty estimation method via contrastive decoding [5]. Using this measurement as a signal, we develop a **dynamic reasoning strategy** that allows us to adaptively adjust the thinking speed during inference, accelerating through straightforward segments while slowing down for difficult reasoning. This strategy not only improves the base LRMs' performance in terms of both efficiency and accuracy, but also highlights the potential paths for thinking speed control methods for future LRM reasoning enhancements.

To summarize, our contributions include:

- We identify and characterize an intrinsic switching mechanism between slow- and fast-thinking modes in LRMs, revealing their native capacity for speed adaptation (Section 2).
- We develop the first reasoning speed control method for LRMs, which demonstrates a promising test-time scaling effect that enables either accelerated reasoning speed or improved accuracy, depending on user-defined requirements (Section 3).
- We propose the first dynamic reasoning strategy that adaptively adjusts the thinking speed of LRMs during inference, achieving both efficiency and accuracy (Section 4).

Notably, all our representation editing algorithms and experiments are implemented using the **vLLM** framework [15]. Benefiting from this foundation and the plug-in design of our method, this inference-time technique can be seamlessly integrated into existing LLM deployment systems. Our method is expected to support broader applications and inspire further research.

## 2 An intrinsic switch between fast- and slow-thinking

We first demonstrate that some LRMs inherently think in both fast- and slow-thinking modes, and there exists a switch to flip between these modes. We study the thought processes (*i.e.*, the text within "<think>" and "</think>") produced by **DeepSeek-R1-Distill-Qwen-7B** [7] on the **MATH-500** [8]

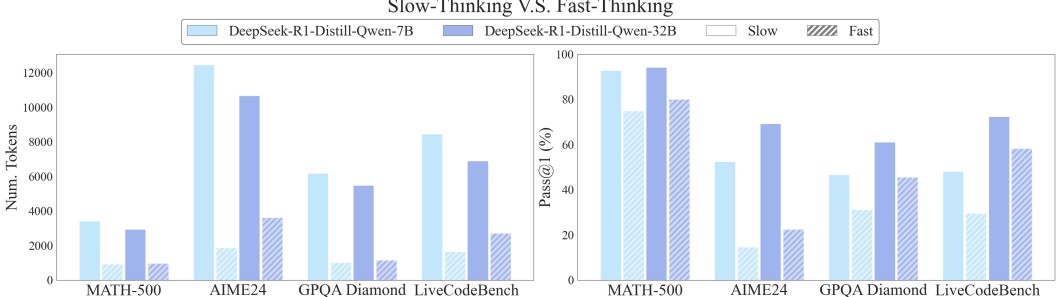

Figure 3: Comparison of LRMs' performances with and without leading words restriction. Initiating thought with "To" significantly reduces token usage for LRMs (**4.4x reduction**) while maintaining comparable performance to regular responses, given the substantial token reduction.

benchmark. Specifically, we analyze the leading word frequencies in the top-100 shortest responses (average length: 262.5 tokens) and the top-100 longest responses (average length: 24,389.2 tokens). As shown in Figure 2, the shortest responses consistently start with leading words like "To" or "First," while the longest responses begin with "Okay" and "Alright." Given this observation, we hypothesize that different leading words at the beginning of LRMs' thought processes might determine the length of their reasoning. To verify this, we use the following prompt template that forces LRMs to start their reasoning with the most common opening words from their shortest responses. We then observe and evaluate how LRMs' thought processes are affected by the restriction of the leading words:

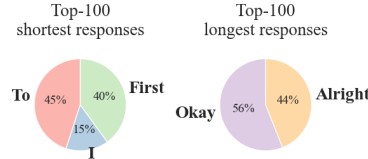

Figure 2: Leading words statistics on MATH-500 from responses of DeepSeek-R1-Distill-Qwen-7B.

```
<|User|>[instruction]<|Assistant|><think>\nTo
```

Compared to the official recommended template, the only modification we make is appending an additional "To" after the thought-begin symbol. Our experimental results show that this minor change leads to a substantial transformation in the output length of our testing LRMs.

**Experimental settings**  We experiment with 2 widely-used LRMs, namely **DeepSeek-R1-Distill-Qwen-7B** and **DeepSeek-R1-Distill-Qwen-32B** [7]. We evaluate these models on **AIME24** [21], **MATH-500** [18], **GPQA Diamond** [27] and **LiveCodeBench** (release_v2) [12]. These benchmarks cover various reasoning skills across multiple disciplines, including math, biology, physics, chemistry, and coding. For each question, we sample responses from LRMs using both the default chat template (with no leading words restriction) and our testing template. For all models, the maximum generation length is set to 32,768 tokens. During sampling, we use a temperature of 0.6, a top-p value of 0.95, and generate 8 responses per query to calculate Pass@1 (*i.e.*, accuracy).

**Results and analysis**  The quantitative comparison between LRM responses under different leading words is shown in Figure 3, including both the number of tokens per response and the Pass@1 performance on the test benchmarks. We notice that: (1) The token consumption of responses starting with "To" is **significantly lower** than that of the regular template, achieving **5.4×** compression for the 7B model and **3.3×** for the 32B model (averaged across benchmarks). (2) Given the substantial token reduction, the shorter responses still preserve 60% (7B) and 68% (32B) of the original model's accuracy on reasoning-intensive tasks, demonstrating their potential for answering simple or routine requests. An illustration of LRM responses under different leading words is provided in Figure 4. The thought process initiated by "To" exhibits vanilla chain-of-thought (CoT) [26, 40] produced by chat models, with a direct and linear reasoning style analogous to System 1 thinking. In contrast, the standard LRM output encompasses comprehensive planning and iterative verification, characteristic of System 2 thinking. Based on the succinctness of language and the efficiency in deriving intermediate results and the final answer, we term the first reasoning style as **fast-thinking** and the latter as **slow-thinking**. Additional case studies on these two thinking modes are provided in Appendix D.1. The intrinsic presence of both System 1 and System 2 thinking modes in LRMs motivates our development of a more flexible thinking speed control method in the following section.

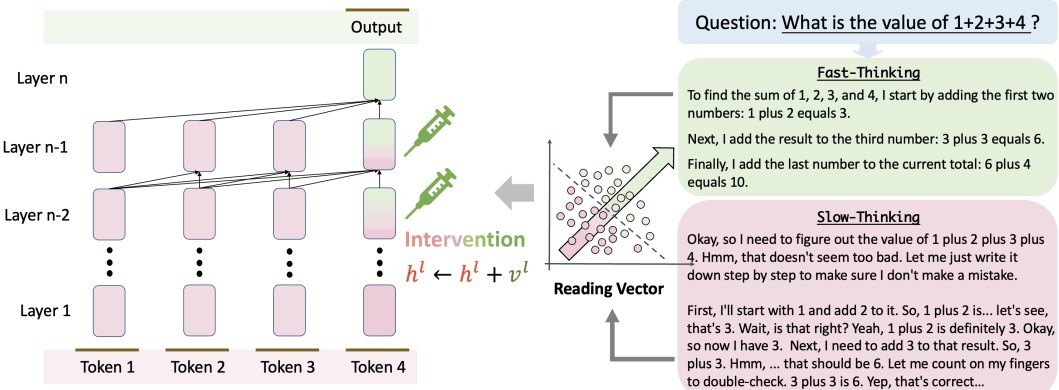

Figure 4: Illustration of our representation engineering process. We extract the directional vector corresponding to the transition from slow-thinking to fast-thinking in the representation spaces of LRMs by contrasting fast and slow thoughts. During inference, we strategically inject this vector to manipulate the model's thinking behavior.

## 3 Thinking speed control via representation engineering

To resemble human cognitive activity, we need more flexible control for transitioning between System 1 and System 2 thinking. Such nuanced control would enable more options for efficiency-accuracy trade-offs and enhance reasoning depth beyond current LRM capabilities. We term this approach **"thinking speed control"**, as it generally affects the reasoning style of LRMs' thought processes and thus the efficiency in deriving answers. To achieve this, we adopt the representation engineering framework (RopE) [45] to identify the directional vector representing the transition between fast- and slow-thinking modes in LRMs' representation spaces [2]. Specifically, our control framework can be decomposed into two key steps (Figure 4): (1) **reading** the direction pointing to the desired thinking mode in the representation space, and (2) **controlling** the LRM's neural activations during inference.

### 3.1 Representation reading

RopE posits that abstract cognitive functions are encoded as linear directions in LLMs' representation space. We hypothesize that distinct reasoning styles are also organized along separate directional subspaces. To compute the vector governing the transition from slow-thinking to fast-thinking, we (i) design stimuli to elicit each thinking mode, (ii) collect hidden representations, and (iii) compute contrasting vectors and extract the first principal component using Principal Component Analysis.

**Step 1: designing stimuli** The stimuli should comprise both positive and negative stimuli to elicit opposing behaviors from the LRM. Given an input prompt $q_i$, we choose the fast response as the positive stimulus, denoted as $T_i^+ = (q_i, a_i^f)$. The corresponding negative stimulus is the same prompt paired with the slow response, denoted as $T^- = (q_i, a_i^s)$. An example pair of fast- and slow-thinking stimuli is shown in Figure 4. Following [45], we retain the initial segment of both fast and slow thinking processes to create the input stimuli. More details are provided in Appendix A.1.

**Step 2: collecting hidden representations** We extract the target LRM's hidden states from each stimulus pair. Specifically, given a layer $l$ and a stimulus pair $S_i = (T_i^+, T_i^-)$, we process each input stimulus through the model and collect the hidden states $(h_{i+}^l, h_{i-}^l)$ at the final token position of the positive and negative stimuli, respectively.

**Step 3: constructing the PCA model** Following [45], we employ a fully unsupervised approach to compute the reading vector corresponding to the transitions from fast- to slow-thinking modes across all stimulus pairs. Denote the hidden state pairs obtained in Step 2 as:

$$\{(h_{1+}, h_{1-}), (h_{2+}, h_{2-}), \cdots (h_{n+}, h_{n-})\}$$

where layer superscript $l$ is omitted for clarity. For half of the hidden state pairs, we compute the difference vector within each pair as $d_i^{(-\rightarrow+)} = h_{i+} - h_{i-}$. For the remaining half of the pairs, we

compute the reversed difference $d_j^{(+\rightarrow-)} = h_{j^-} - h_{j^+}$. We then perform PCA on the combined set of both difference directions $\{d_i^{(-\rightarrow+)}\} \cup \{d_j^{(+\rightarrow-)}\}$. The first principal component $v$ from this PCA optimally aligns with our desired slow→fast direction (*i.e.*, $v^T d_i^{(-\rightarrow+)} > 0$) and contrasts with the reversed direction (*i.e.*, $v^T d_j^{(+\rightarrow-)} < 0$). This component thus serves as our reading vector $v^l$ for layer $l$. More details can be found in Appendix A.2.

## 3.2 Representation controlling

After obtaining the contrastive direction vector $v$, we can steer LRM's reasoning styles by injecting this vector into its hidden states during inference. As shown in Figure 4, specifically, for each target intervention layer $l \in L$, we modify the hidden state $h^l$ with a steering intensity $\alpha$:

$$h^l \leftarrow h^l + \alpha \cdot v^l \tag{1}$$

Unless otherwise specified, we apply this intervention during the generation of every token in the response. More details can be found in Appendix A.3. Ideally, an effective control should produce:

- Shorter and more concise response when $\alpha > 0$ (enhancing fast-thinking);
- More complex and deliberate reasoning when $\alpha < 0$ (encouraging slow-thinking).

The following experimental results demonstrate the success of our controlling method, which exhibits an inspiring **scaling effect**. As the length of the reasoning increases (*i.e.*, the slower LRMs think), the final performance on various reasoning benchmarks improves accordingly.

## 3.3 Experimental results and analysis

**Experimental settings**  We conduct the experiments on 4 LRMs: **DeepSeek-R1-Distill-Qwen-7B**, **DeepSeek-R1-Distill-Qwen-32B**, **QwQ-32B** [34] and **Qwen3-8B** [33]. For representation reading, we use **only** the MATH training set (7.5k math problems) to sample both fast and slow responses from the LRMs. Then we utilize the sampled responses to construct stimulus pairs and collect layer-wise hidden states from each model. Finally, we compute the reading vectors for each LRM via PCA.

To test the effect of our reading vectors, we apply the control to the LRMs on **AIME24**, **MATH-500**, **GPQA Diamond**, and **LiveCodeBench**, which cover math reasoning, biology, physics, chemistry, and coding. We sweep the steering intensity $\alpha$ in Equation (1) across positive and negative values to evaluate the LRMs' performances when their thought processes are **accelerated** or **decelerated**.

For comparison, we utilize the following 2 baselines:

- **Budget Forcing** [25, 43]: Forcing the LRM to early-exit its reasoning by appending the string "Final Answer:" to its current trace to prompt the final answer.[2] We align the early exit position to the response length of each LRM under different $\alpha > 0$ values for comparison.
- **Thought Extrapolation** [25]: Extending the reasoning trace by appending the string "Wait" to the LRM's solution, encouraging the model to reflect on its current answer. We run this baseline by recursively appending 1x/2x/3x times of "Wait" to the LRMs' last generations.

For all experiments, we use **vLLM** [15] and set the maximum generation length to 32,768 tokens. To avoid randomness, we generate 8 responses per query, with the temperature set to 0.6 and a top-p value of 0.95, as officially recommended [7]. The evaluation code is adapted from Sky-T1 [32]. More details can be found in Appendix A.4. We run all the experiments with NVIDIA A100 80GB GPUs.

**Results and analysis**  A series of responses generated by our test model under different steering intensities are shown in Appendix D.2. We observe a clear spectrum of output styles as $\alpha$ changes. When $\alpha > 0$ increases, the reasoning steps become increasingly intuitive, featuring straightforward step-by-step derivations with structured mathematical expressions. Conversely, as $\alpha < 0$ decreases, each reasoning step becomes more deliberate and careful, with an increasing amount of validations and backtrack strategies. Such reasoning style changes align with our expectations of the thinking speed control method. The quantitative performances shown in Figure 5 exhibit a clear **scaling effect**:

---

[2]Note that budget forcing works only for questions whose final answers can be expressed using a few tokens, such as numbers or options. Therefore, this baseline is omitted on LiveCodeBench, as it is a coding benchmark.

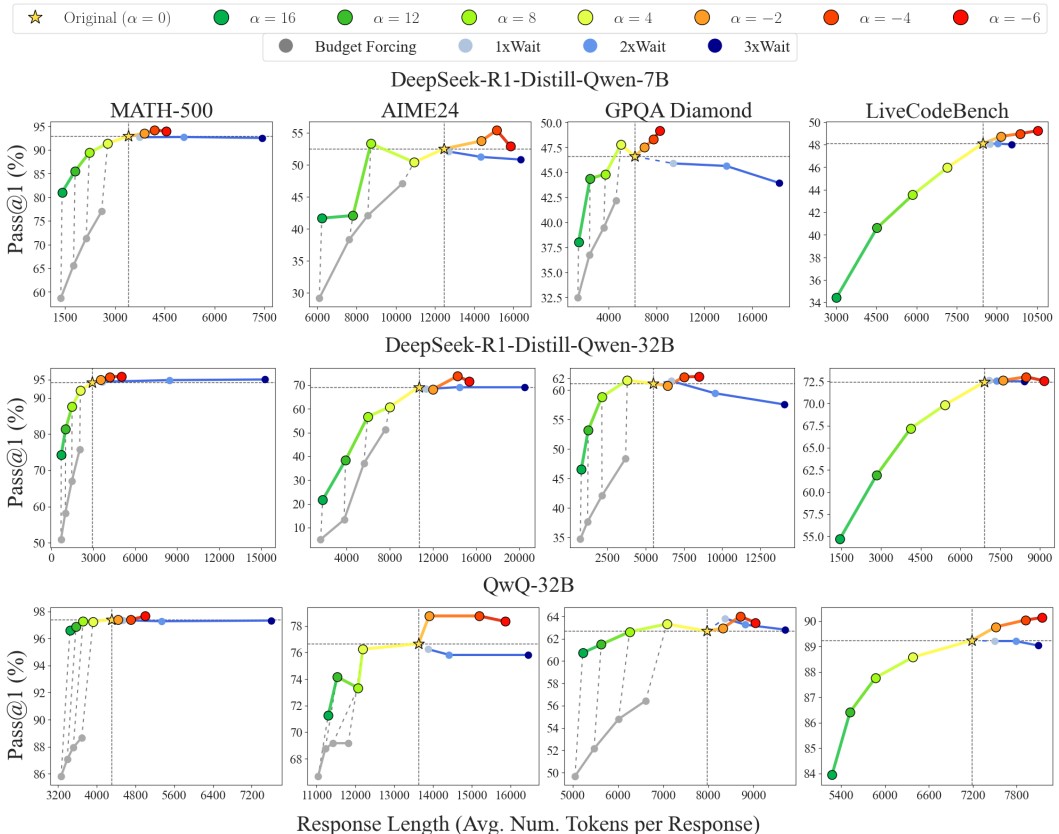

Figure 5: Scaling effects of thinking speed control. We show the trade-off between response length (x-axis, average token count) and reasoning accuracy (y-axis, Pass@1). Key annotations: (1)"⋆": Baseline model performance. (2)"●∼●": Representation control with different steering intensity $\alpha$ (positive to negative). (3)"●": Budget-forced early exiting at varying positions. (4)"●∼●": Thought extrapolation by appending 1x/2x/3x times of "Wait". Our control method consistently shows superior performance compared to baselines. The table version of the above results is presented in Table A4.

all LRMs' performances improve with increased response lengths, ultimately exceeding their original performance through our control method. The results on Qwen3-8B are shown in Figure A3. When accelerating thinking ($\alpha = 16, 12, 8, 4$), we observe clear advantages over the early-exiting baselines, with averaged Pass@1 improvements of $+11.4/11.2/10.7/8.2$ respectively. We further demonstrate the superiority of our acceleration method when combined with parallel search in Appendix C. Slowing down the thinking speed generally yields higher accuracy than the original responses, with Pass@1 improvements of $+0.51/1.46/1.17$ for $\alpha = -2/-4/-6$, averaged over all models and benchmarks. It also clearly outperforms prompt-based thought extrapolation baselines, which can even degrade performance as reflection time increases. We also notice that different LRMs have varying sensitivities to both representation- and prompt-based control (*e.g.*, thought extrapolation). This results in uneven changes in the response lengths of different models as $\alpha$ or reflection time varies. We leave the development of more uniform control over the response length for future work.

The success of our control method stems from two key advantages: (1) token-level granularity and (2) representation-based operation. The token-level control enables precise, continuous steering of reasoning styles throughout the generation. By working directly with the internal representations instead of the reasoning text, our approach preserves the natural reasoning flow, better leveraging LRMs' inherent capabilities and thus surpassing human-crafted prompt design baselines.

## 4 Adaptive control of thinking speed

Our thinking speed control experiments show that LRMs' reasoning processes can be either accelerated for faster responses or decelerated for deeper thought through constant steering. However, the

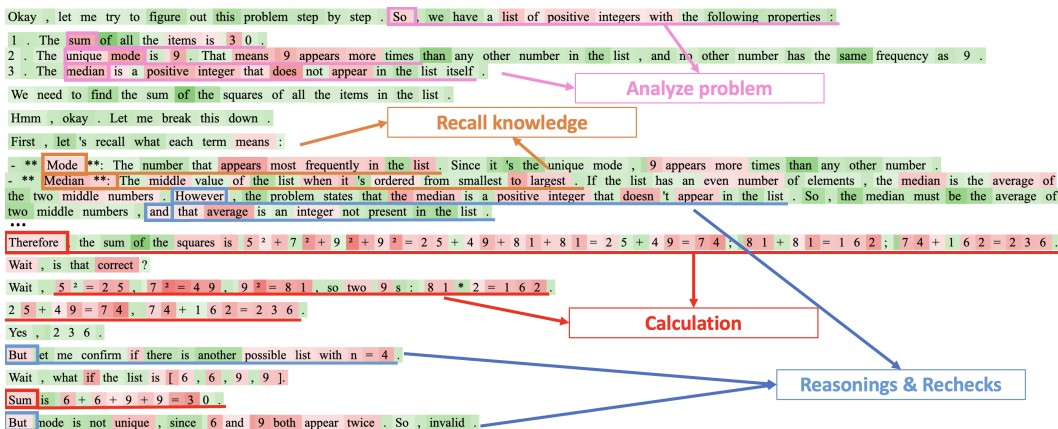

Figure 6: Visualization of our computed reasoning difficulty in DeepSeek-R1-Distill-Qwen-7B outputs. We highlight tokens with high reasoning difficulty defined in Equation (4) using "███", with **darker shades** indicating **higher values**. Tokens with lower reasoning difficulty are highlighted using "███" with **darker shades** indicating **lower values**.

thinking speed of real humans varies throughout the reasoning process. We quickly **skim over the straightforward parts** of the solution and spend more time **deliberating on the challenging parts** in the reasoning. To better approximate human-like reasoning, we propose a dynamic reasoning strategy that **adaptively adjusts** the thinking speed within one single reasoning trace to achieve both accuracy and efficiency. As a pioneering work, we hope our method will inspire future research into inference-time thinking control, leading to enhanced reasoning performances of current LRMs.

## 4.1 Adaptive speed control via difficulty estimation

To balance between efficiency and accuracy, we need to find a real-time signal (*e.g.*, a "traffic light") during inference that indicates when it is safe to process quickly and when it is necessary to slow down and think more carefully. LLMs have been shown to primarily process high-level semantics in later layers, resulting in high logit variations between early and late layer predictions when processing complex information [5, 23]. Based on this observation, we hypothesize that reasoning involving high difficulty or complex strategies that enhance LRMs' accuracy, such as reflections and analysis, should also involve high-level semantic processing. Such features would make difficult reasoning segments detectable through logit variation patterns. Given a token sequence $\{x_1, x_2, \cdots x_{t-1}\}$ and an $N$-layer transformer, we denote the output of the $l$-th layer as $h_t^l$. The vocabulary head $\phi(\cdot)$ computes the next-token distribution:

$$p(x_t|x < t) = \text{softmax}(\phi(h_t^l)) \tag{2}$$

For a defined set of early layers $L \subset \{1, \cdots, N-1\}$, we measure the difference between each early-layer distribution $p_l(\cdot|x_{<t})$ and the final layer's $p_N(\cdot|x_{<t})$ using Jensen-Shannon divergence:

$$D\left(p_N(\cdot|x_{<t}), p_l(\cdot|x_{<t})\right) = \text{JSD}\left(p_N(\cdot|x_{<t})||p_l(\cdot|x_{<t})\right). \tag{3}$$

We use the average divergence across early layers to quantify the reasoning difficulty at token $x_t$:

$$d(x_t) = \underset{l \in L}{\text{avg}}\ D\left(p_N(\cdot|x_{<t}), p_l(\cdot|x_{<t})\right) \tag{4}$$

Higher values of $d(x_t)$ should indicate greater reasoning difficulty at position $t$. To validate our hypothesis, we first identify the top 100 tokens with the highest logit variations between the first half of the LRM's layers and the last layer in the math reasoning outputs. We categorize these tokens based on their

Table 1: (Part of) Top 100 tokens with high variations in logits obtained from early and late layers.

| Category | Top Tokens |
| --- | --- |
| Reflections | Wait, Alternatively, However, mistakes, ... |
| Calculations | Equation, equals, multiply, approx, ... |
| Analysis | sequentially, What, need, analysis, ... |

most-related reasoning behaviors in Table 1. We also visualize the reasoning difficulty distribution using Equation (4) in Figure 6. Transitions from low-difficulty (green) to high-difficulty (red) regions

Table 2: Results from our adaptive speed control experiments. Our adaptive control method (highlighted by ▒) consistently outperforms the original LRMs in both accuracy and token usage.

| Methods | Performance (Pass@1 (%)↑ / Num. Tokens ↓) | | | |
| --- | --- | --- | --- | --- |
| | **MATH-500** | **AIME24** | **AIME25** | **GPQA Diamond** |
| **DeepSeek-R1-Distill-Qwen-7B** | 92.9 / 3403.9 | 52.5 / 12451.2 | 40.0 / 13688.9 | 46.6 / 6189.3 |
| *w.* 1xWait | 92.7 / 3744.2 | 52.1 / 12704.0 | 39.6 / 13866.6 | 45.9 / 9376.0 |
| *w.* Constant steering $\alpha = 4$ | 91.4 / 2784.9 | 50.4 / 10942.7 | 37.9 / 12315.4 | 47.8 / 5016.2 |
| *w.* Constant steering $\alpha = -4$ | 94.1 / 4193.6 | 55.4 / 15144.8 | 39.6 / 16168.9 | 48.3 / 7732.3 |
| *w.* Adaptive control | **93.7 / 3122.8** | **53.8 / 10850.9** | **42.3 / 12379.5** | **48.8 / 5421.6** |
| **DeepSeek-R1-Distill-Qwen-32B** | 94.2 / 2947.3 | 69.2 / 10679.6 | 51.7 / 12853.1 | 61.1 / 5493.1 |
| *w.* 1xWait | 94.4 / 3634.4 | 68.3 / 11323.3 | 51.2 / 13121.1 | 61.5 / 6671.6 |
| *w.* Constant steering $\alpha = 4$ | 92.0 / 2088.7 | 60.8 / 8000.1 | 43.8 / 9496.0 | 61.2 / 3793.4 |
| *w.* Constant steering $\alpha = -4$ | 95.8 / 4199.4 | 73.8 / 14254.7 | 55.4 / 16108.7 | 62.2 / 7500.3 |
| *w.* Adaptive control | **94.9 / 2732.9** | **70.7 / 9322.5** | **51.9 / 10542.0** | **61.5 / 4774.6** |
| **QwQ-32B** | 97.4 / 4305.0 | 76.7 / 13627.3 | 65.8 / 15852.1 | 62.7 / 7968.7 |
| *w.* 1xWait | 97.3 / 4540.5 | 76.2 / 13868.8 | 64.6 / 16010.0 | 63.8 / 8369.2 |
| *w.* Constant steering $\alpha = 4$ | 97.2 / 3921.4 | 76.2 / 12191.6 | 67.5 / 14495.1 | 63.3 / 7083.8 |
| *w.* Constant steering $\alpha = -4$ | 97.4 / 4704.1 | 78.7 / 15191.7 | 71.2 / 17255.8 | 64.0 / 8707.9 |
| *w.* Adaptive control | **97.4 / 4133.6** | **77.8 / 12364.7** | **67.4 / 15150.1** | **64.1 / 7639.1** |
| **Qwen3-8B** | 96.8 / 5456.2 | 75.0 / 14753.6 | 62.9 / 17797.0 | 60.1 / 8378.9 |
| *w.* 1xWait | 96.6 / 5662.0 | 74.2 / 14874.7 | 62.5 / 17909.2 | 59.8 / 8489.2 |
| *w.* Constant steering $\alpha = 4$ | 97.1 / 5030.9 | 73.8 / 14131.4 | 63.3 / 16695.2 | 60.3 / 7615.0 |
| *w.* Constant steering $\alpha = -4$ | 97.4 / 5878.3 | 75.4 / 15910.2 | 66.2 / 18628.4 | 61.0 / 9168.5 |
| *w.* Adaptive control | **97.1 / 5170.8** | **77.5 / 13629.0** | **65.4 / 17410.9** | **61.2 / 7894.0** |

usually indicate the start of (1) problem analysis, (2) knowledge retrieval, (3) numerical computation, and (4) logical deduction. This pattern closely mirrors human problem-solving processes, making it an effective natural signal for switching to slow-thinking mode when careful processing is required in the subsequent reasoning. Based on these observations, we develop a sliding-window algorithm to monitor LRMs' reasoning difficulties and dynamically adjust the steering intensity during inference time. We track the reasoning difficulties of the most recent $k$ tokens in a continuously updated window, denoted as $W \in \mathbb{R}^k$. At each generation step, we compare the current token's difficulty against a dynamic threshold:

$$\text{threshold} = \mu_W + \lambda \cdot \sigma_W \tag{5}$$

where $\mu_W, \sigma_W$ represent the window's mean and standard deviation, respectively, and $\lambda$ serves as the outlier detection parameter. When encountering significantly higher difficulty (signaling important reasoning segments), we slow down the thinking speed for deeper analysis (similar to stepping on the brake). Otherwise, we progressively increase the steering intensity until we reach the preset upper bound. A detailed pseudocode can be found in Algorithm 1.

## 4.2 Experimental results and analysis

**Experimental settings** We conduct the experiments on **DeepSeek-R1-Distill-Qwen-7B**, **DeepSeek-R1-Distill-Qwen-32B**, **QwQ-32B** and **Qwen3-8B**. We set the sliding window size as $k = 8$ tokens. The outlier detection threshold $\lambda$ is set to 2.0 for the 7B and 8B models and 1.5 for the 32B model. The steering intensity $\alpha$ is constrained to the range $[-4, 4]$. To decide the range of early layers for logits contrasting, we partition the transformer layers into buckets and select the optimal one based on validation results. Following [5], we set 2 buckets $[1, 14]$ and $[14, 27]$ for the 7B model (28 layers),

2 buckets $[1, 18]$ and $[18, 35]$ for the 8B model (36 layers), and 3 buckets $[1, 21]$, $[22, 42]$ and $[43, 63]$ for the 32B models (64 layers). To determine the optimal bucket for math reasoning, we evaluate model performance across different buckets using the **AMC** problem set [17] as validation data. The validation results and more details on our parameter settings can be found in Appendix B. For math reasoning tests, we use **MATH-500**, **AIME24** and **AIME25** [22]. **GPQA Diamond** is also included as a generalization test. The vLLM generation parameters remain the same as in Section 3.3.

**Results and analysis**    The results from our adaptive speed control experiments are shown in Table 2. Compared to each LRM's original performance and the thought extrapolation baseline, our adaptive control method consistently shows advantages in terms of both Pass@1 accuracy and token usage (+1.26% accuracy while -8.56% token usage, averaged across all models and benchmarks). For a comprehensive comparison, we include constant steering results at boundary intensities $\alpha = 4$ and $-4$, which collectively validate our approach's ability to combine the efficiency of fast thinking with the accuracy of slow reasoning. For a straightforward demonstration of our control method's effectiveness, we provide detailed case studies and quantitative analysis in Appendices B.6 and B.7. These examples illustrate how our strategic speed adjustments guide LRMs to optimally leverage their reasoning capabilities, yielding more intelligent outputs. The complete ablation studies on the parameter configurations and the effect of our reasoning difficulty measurement are provided in Appendix B.

## 5    Related Works

**Large reasoning models**    Large Language Models (LLMs) like GPT-4o [10] and DeepSeek V3 [6] typically employ fast, intuitive, and organized reasoning, which limits their ability to self-criticize and self-correct [9, 14] and their performance on advanced reasoning tasks. To address these limitations, a new series of models, named Large Reasoning Models (LRMs), has emerged, including OpenAI o1 series [11] and DeepSeek R1 series [7]. Compared to LLMs, LRMs employ extended reasoning chains by incorporating self-verification mechanisms and trial-and-error exploration. This slow-thinking paradigm enables LRMs to achieve substantial breakthroughs in complex reasoning tasks.

**Reasoning strategy**    Recent research has developed various inference-time strategies to enhance LRMs' efficiency and effectiveness. One direct way to balance accuracy and efficiency is to explicitly instruct LRMs to reason within predefined token budgets [35, 1, 25]. For example, one can either force LRMs to stop reasoning and directly give the answer at the given budget [25, 43] or ask LRMs to self-reflect on the original responses for better performance [44, 41]. However, studies have pointed out the insensitivity of current LRMs to time constraints in prompts [31, 16]. Our experiments in Section 3.3 also show the suboptimality of prompt-based methods. Other approaches propose dynamically routing inputs based on task difficulty to different models [30, 28, 36], which usually requires the deployment of multiple LRMs. Another line of work attempts to increase the width of the solution search space (i.e., parallel search) [20, 29], including majority voting, self-consistency [37], and Best-of-N [18], often with the assistance of an external verifier for quality assessment.

Our representation editing-based method maintains full orthogonality to current techniques, which makes our approach particularly suitable for integration with the above methods. An empirical combination of our approach with parallel search is provided in Appendix C. We anticipate fruitful combinations of our work with existing techniques in future research.

## 6    Conclusion

This work rethinks dual-process reasoning (System 1 vs. System 2) in Large Reasoning Models (LRMs). To approximate human intelligence, we propose an inference-time method that enables dynamic thinking speed adjustment in LRMs, optimizing efficiency-accuracy trade-offs. We first reveal that some LRMs intrinsically support both thinking modes through an easy switching mechanism. We then extract the steering vector from LRMs' representation space that controls the thinking speed, which enables: (1) fast yet accurate responses, or (2) enhanced accuracy through slow reasoning. Finally, we utilize real-time difficulty estimation as the control signal, enabling fast processing of easy steps and deeper analysis on complex reasoning. Combining these techniques, our reasoning strategy shows consistent improvements in both accuracy and token usage across multiple LRMs and

advanced benchmarks. We hope this work can provide new insights for future developments in LRM reasoning strategies.

# 7 Limitation and future work

While our study provides valuable insights into the thinking abilities of Large Reasoning Models (LRMs), it also has several limitations. First, our experiments reveal that the fast-thinking abilities of LRMs can be triggered by certain opening words. The underlying reasons for this intriguing phenomenon remain unclear. We leave a more systematic and thorough investigation of this phenomenon for future work.

Second, in our thinking speed control experiment in Section 3.3, we used the same steering intensities for all LRMs and benchmarks. While this approach generally yields effective results, we also observe that different LRMs have varying sensitivities to the steering vectors, reflected by the uneven changes in the response lengths across different steering intensities. Due to these inconsistencies, a model-agnostic approach to determining steering intensities may be needed for more controllable reasoning.

Finally, our adaptive control implementation employs a heuristic sliding-window algorithm for steering intensity adjustment. Although this method demonstrates effectiveness across all tested models and benchmarks, we believe there is room for further improvement through more intricate approaches. A promising enhancement would involve developing an integrated framework that directly computes the optimal steering intensities based on real-time difficulty estimates, rather than relying on heuristic adjustments. We believe that such an organic coupling of difficulty estimation with speed adjustments could further optimize the efficiency-accuracy trade-off, representing a valuable direction for future investigation.

# 8 Societal impact discussion

Our research on thinking speed control in large reasoning models (LRMs) offers several positive societal impacts. As an inference-time technique, it empowers users with flexible control over LRMs' reasoning processes, enabling: (1) faster responses for routine queries, and (2) deeper analysis for advanced reasoning tasks when accuracy is prioritized. Our adaptive speed adjustment algorithm further optimizes the efficiency-accuracy tradeoff, surpassing current LRM capabilities. We believe our work would inspire future research on the development of more efficient and intelligent AI systems.

We foresee no negative societal impacts from our research, as our work focuses exclusively on enhancing reasoning performance in existing models. All data used in our experiments are sourced from public datasets and contain no harmful or sensitive content.

## Acknowledgment

This work was supported in part by The National Nature Science Foundation of China (Grant No: 62303406, 62273302, 62036009), in part by Zhiyuan Laboratory (NO. ZYL2024022b).

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

# Appendices

# A Supplementary materials for Section 3

## A.1 Details on stimuli design

As described in Section 3, we use both fast- and slow-thinking responses as the positive and negative stimulus. The 7.5k questions from the MATH [8] training set are utilized to sample responses of these two modes from the experimental LRMs. We first select those pairs that both end with correct answers. For each response pair, we retain only the initial segment of the thought process to construct the stimuli. We do this by:

1. Extracting the thought process (*i.e.*, the reasoning chain between "<think>" and "</think>") and using double line breaks (*i.e.*, "\n\n") to segment each steps within.
2. For fast-thinking response, we retain only the first 2 steps in its original thought process as the positive stimulus.
3. For slow-thinking response, we truncate it to the step that has the nearest response length compared to its paired positive stimulus.

An example stimulus pair is shown in Figure A1. Following [45], we retain the initial segment of the reasoning chain in our stimuli for three key reasons:

- To capture meaningful differences in reasoning conciseness, we avoid using only the first token's representation (e.g., opening words like "To" or "Alright"), as this would reduce the contrast to mere differences in word embeddings. By including part of the reasoning process, we ensure the resulting steering vector shares dimensional features with the representations of other reasoning tokens in the thought process, which would minimize the unnecessary disruption during inference-time control.
- Since our representation control applies to every token during inference, we need to ensure that the extracted direction vector contains semantic features that encourage the model to continue reasoning rather than terminate prematurely. Therefore, we choose to use only the initial part of the thought process instead of the entire process to avoid signals like end-of-sequence or end-of-thought tokens, as the latter might cause the controlled LRM to end the reasoning process prematurely.
- To cancel out positional embedding information from the semantics of the representation, we need to ensure that the contrasting stimuli share approximately the same length. Additionally, since fast responses are typically shorter than their slow counterparts, using complete fast responses would require truncating slow responses at comparable lengths, which could potentially cut them during early or mid-reasoning stages. This could introduce noise into the contrasting vectors, as they would capture differences in reasoning stages rather than purely succinctness-related information. These considerations further demonstrate why using entire thought processes for stimulus construction would be suboptimal.

Since the average number of steps in the fast-thinking response is 7.1, we decide to truncate each fast-thinking response after the first 2 steps. We also compare the steering performance by taking representations from different positions in the stimuli, as shown in Table A1. The results indicate that using the representation from the last token of the initial segment generally yields the best performance.

Table A1: Steering performance using token representations extracted from different positions in the stimulus. The evaluation is conducted on DeepSeek-R1-Distill-Qwen-7B using the GPQA Diamond benchmark. **Baseline performance ($\alpha = 0$): 46.6% / 6189.3 (Pass@1 / Num. Tokens).** Our default configuration ( ) generally achieves the best performance across different steering intensities.

| Representation Position | GPQA Diamond Performance (Pass@1 (%)↑ / Num. Tokens ↓) | | | |
|---|---|---|---|---|
| | $\alpha = -6$ | $\alpha = -4$ | $\alpha = 4$ | $\alpha = 8$ |
| **End of the initial segment** | **49.2** / 8277.0 | **48.3** / 7732.3 | **47.8** / 5016.2 | 44.8 / 3712.3 |
| **First token** | 47.9 / 7520.7 | 47.5 / 7093.4 | 46.7 / 5692.9 | **47.2** / 4871.9 |
| **End of the entire thought** | 48.3 / 7205.3 | 47.0 / 6660.6 | 47.7 / 5955.4 | 46.1 / 5326.3 |

**Question**

What is the greatest integer $x$ such that $|6x^2 - 47x + 15|$ is prime?

**Fast-Thinking**

To determine the greatest integer $x$ such that $|6x^2 - 47x + 15|$ is a prime number, I first need to analyze the quadratic expression inside the absolute value.

I'll start by factoring the quadratic equation $(6x^2 - 47x + 15)$. To do this, I'll look for two numbers that multiply to $6 \times 15 = 90$ and add up to -47. The numbers -45 and -2 fit these criteria because $-45 \times (-2) = 90$ and $-45 + (-2) = -47$).

**Slow-Thinking**

Okay, so I have this problem here: I need to find the greatest integer $x$ such that the absolute value of $6x^2 - 47x + 15$ is a prime number. Hmm, prime numbers are numbers greater than 1 that have no positive divisors other than 1 and themselves. So, I need to find $x$ such that $|6x^2 - 47x + 15|$ is prime. And specifically, I'm looking for the greatest such integer $x$.

First, let me consider the quadratic expression $6x^2 - 47x + 15$. Since it's a quadratic, it can be factored or maybe I can find its roots to understand its behavior. Maybe factoring will help me see if the quadratic can be broken down into simpler terms, which might make it easier to analyze.

Figure A1: An example of a positive and negative stimulus pair sampled from the MATH training set.

## A.2 Details on PCA model

After filtering out incorrect responses sampled from the MATH training set, we used approximately 6k stimulus pairs for representation collection. We randomly chose 4k pairs to compute the difference vectors, with half of the pairs to compute directional vectors $d_i^{(-\rightarrow+)}$ and the other half to compute the reversed-direction vectors $d_j^{(+\rightarrow-)}$. We then applied PCA on the combined set $\{d_i^{(-\rightarrow+)}\} \cup \{d_j^{(+\rightarrow-)}\}$ to extract the first principal component that best separates these 2 oppositely-directional vector groups. Additionally, since the resulting PCA component lacks a well-defined direction (*i.e.*, it may assign positive scores to $\{d_i^{(-\rightarrow+)}\}$ and negative scores to $\{d_j^{(+\rightarrow-)}\}$, or vice versa), we calibrate its direction to ensure our final steering vector consistently points from slow- to fast-thinking representations. This calibration is achieved by computing the mean score across $\{d_i^{(-\rightarrow+)}\}$. If the average score is positive, we directly use the first principal component as our steering vector; otherwise, we invert it by multiplying by $-1$.

The remaining 2k pairs from the MATH set served as a validation set, where we calculated difference vectors in both directions to verify whether the principal component could effectively separate these vectors based on their directions. The validation results, shown in Figure A2, confirm both the success of our extraction method and the presence of response succinctness-related information in the models' hidden states.

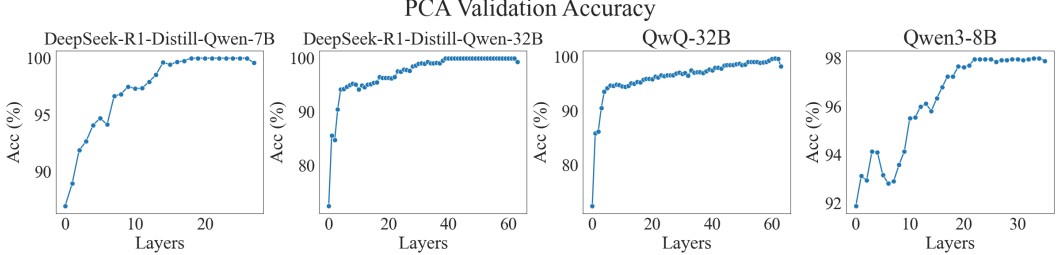

Figure A2: The classification accuracy of the PCA model for each LRM on the validation set.

## A.3 Details on representation controlling

In our experiment in Section 3.3, we evaluate steering intensities $\alpha$ from $[16, 12, 8, 4, 0, -2, -4, -6]$. To decide the range of layers to control, we first study the PCA results in Figure A2 and note that the later layers exhibit clearer thinking-mode separation. We hypothesize that higher layers contain stronger encoding of response succinctness, which is also consistent with established findings about

higher layers processing semantic information [23, 24]. Furthermore, we determine the control range for each of our testing models by:

1. Selecting layers with PCA validation accuracies close to or equal to 100%, which primarily focuses on the last few layers.
2. Further filtering based on our early observations of their performance on the MATH dataset, mainly determined by the stability of the fluctuations in both response lengths and accuracies under different steering intensities.

Specifically, for:

- **DeepSeek-R1-Distill-Qwen-7B** (28 layers): we control the last 10 layers except for the last layer, namely $l \in [19, 27]$.
- **Qwen3-8B** (36 layers): we control the last 10 layers except for the last layer, namely $l \in [26, 35]$.
- **DeepSeek-R1-Distill-Qwen-32B** (64 layers): we control the last 20 layers except for the last layer, namely $l \in [44, 63]$
- **QwQ-32B** (64 layers): we control the last 15 layers except for the last layer, namely $l \in [49, 63]$.

Each model's final layer is excluded to preserve output distribution stability.

## A.4 Details on experimental settings

**Baselines**   To ensure a fair comparison between our control method and baselines, for the budget forcing method, we constrain the maximum output length for each question to match the average response length achieved under different $\alpha > 0$ steering intensities for the same question. This alignment guarantees equivalent token budgets for both approaches on the test benchmarks. Furthermore, the average output length serves as a prior for the difficulty of the problem, allowing the budget forcing method to exit earlier for easy problems and later for difficult problems, which provides a more rigorous baseline. Following [43], after reaching the maximum output length, we append the string "\n\n**Final Answer**\n\boxed{" to elicit the LRM's early judgment for the final answer. At this step, we set the maximum generation length to 10 to prevent the LRM from continuing to reason from the early exit position. Given this output restriction, we exclude this baseline from coding benchmarks as it may truncate the code generation.

For prompt-based thought extrapolation, we first append an additional "Wait" to the LRM's original response and acquire its new final answer after the first recheck. We then iteratively append "Wait" two and three times to its last generation to obtain the answer after additional rechecks. The maximum model length is set to 32,768 tokens throughout all regenerations.

**Evaluation**   For evaluation of each benchmark, we use the following templates from [32]:

**MATH-500**

```
<|User|>Return your final response within \boxed{}. [question]
<|Assistant|><think>\n
```

**AIME24**

```
<|User|>Return your final response within \boxed{}. [question]
<|Assistant|><think>\n
```

**GPQA Diamond**

```
<|User|>Return your final response within \boxed{} and only include the
letter choice (A, B, C, or D) as your final response.  [question]
<|Assistant|><think>\n
```

### LiveCodeBench

- stdin template:

```
<|User|>Generate an executable Python function generated from the
given prompt.  The function should take stdin as input and print the
output.  Simply call the function after the definition.  [question]
<|Assistant|><think>\n
```

- Non-stdin template:

```
<|User|>Generate an executable Python function generated from the
given prompt.  Return the function body without invoking it at the
final solution.  [question]
<|Assistant|><think>\n
```

## A.5  Results on Qwen3-8B

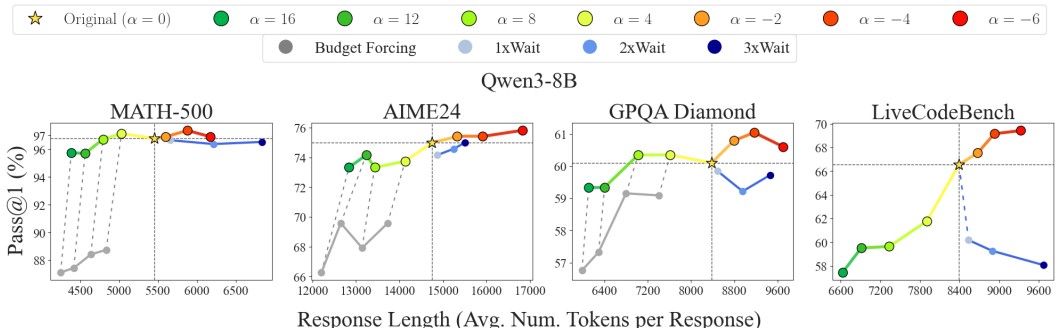

Figure A3: Scaling effects of thinking speed control on Qwen3-8B. We show the trade-off between response length (x-axis, average token count) and reasoning accuracy (y-axis, Pass@1).

We present the results of our thinking speed control experiments on Qwen3-8B in Figure A3 and Table A2. All parameters and generation settings remain consistent with those in Section 3.3. The results align with the scaling effect observed in Figure 5: the LRM's performance improves under slow thinking ($\alpha < 0$), while fast-thinking steering consistently surpasses budget-forced baselines by a clear margin. These findings demonstrate the generalizability of our control method to the latest generation of LRMs.

## A.6  Effects of extreme values of steering intensity $\alpha$

To show the effects of extreme values of steering intensity, we extend our experiments in Figure 5 by scaling up the absolute value of $\alpha$ when controlling the thinking speed of **DeepSeek-R1-Distill-Qwen-7B** on **AIME24**. The results are shown in Table A3. Increasing $\alpha$ leads to progressively shorter responses. And when $\alpha > 0$ becomes too large, we observe repetitive generation behavior. Similarly, for $\alpha < 0$, increasing $|\alpha|$ produces longer responses but also leads to repetition at extreme values. This behavior is expected, as $\alpha$ serves as a hyperparameter and, like many others in machine learning, exhibits an effective operating range beyond which model behavior can become unstable.

## A.7  Supplementary results for Figure 5.

Table A2: Experimental results of our thinking speed control experiment on Qwen3-8B. The figure version is provided in Figure A3. We compute the average performance difference of each setting compared to the original LRM in the last column ($\Delta$ Avg.)

| Methods | Performance (Pass@1 (%)↑ / Num. Tokens ↓) | | | | |
|---|---|---|---|---|---|
| | **MATH-500** | **AIME24** | **GPQA Diamond** | **LiveCodeBench** | **$\Delta$ Avg.** |
| **Qwen3-8B** | | | | | |
| Original | 96.8 / 5456.2 | 75.0 / 14753.6 | 60.1 / 8378.9 | 66.5 / 8391.1 | - / - |
| $\alpha = -6$ | 96.9 / 6177.0 | **75.8** / 16830.2 | 60.6 / 9702.5 | **69.4** / 9326.3 | **+1.09** / +1264.0 |
| $\alpha = -4$ | **97.4** / 5878.3 | 75.4 / 15910.2 | **61.0** / 9168.5 | 69.2 / 8927.4 | **+1.15** / +726.2 |
| $\alpha = -2$ | 96.9 / 5600.9 | 75.4 / 15324.4 | 60.8 / 8794.2 | 67.5 / 8674.3 | **+0.56** / +353.5 |
| $\alpha = 4$ | 97.1 / 5030.9 | 73.8 / 14131.4 | 60.3 / 7615.0 | 61.8 / 7902.1 | -1.35 / -575.1 |
| $\alpha = 8$ | 96.7 / 4792.5 | 73.3 / 13432.2 | 60.3 / 7023.7 | 59.6 / 7332.4 | -2.12 / -1099.8 |
| $\alpha = 12$ | 95.7 / 4562.3 | 74.2 / 13236.4 | 59.3 / 6401.6 | 59.5 / 6910.3 | -2.42 / -1467.3 |
| $\alpha = 16$ | 95.8 / 4384.1 | 73.3 / 12839.8 | 59.3 / 6107.3 | 57.5 / 6627.6 | -3.12 / -1755.3 |
| 1xWait | 96.6 / 5662.0 | 74.2 / 14874.7 | 59.8 / 8489.2 | 60.2 / 8536.4 | -1.90 / +145.6 |
| 2xWait | 96.4 / 6211.9 | 74.6 / 15244.9 | 59.2 / 8954.0 | 59.3 / 8896.8 | -2.22 / +582.0 |
| 3xWait | 96.5 / 6837.7 | 75.0 / 15509.0 | 59.7 / 9459.4 | 58.1 / 9675.3 | -2.27 / +1125.4 |

Table A3: Performance of **DeepSeek-R1-Distill-Qwen-7B** on **AIME24** under extreme steering intensities. The steering effect remains stable and effective across a wide range of $\alpha$; only at extreme values of $|\alpha|$ does the model begin repetitive generation.

| Performance | Steering Intensity $\alpha$ | | | | | | | | | |
|---|---|---|---|---|---|---|---|---|---|---|
| | $\alpha = 64$ | $\alpha = 32$ | $\alpha = 16$ | $\alpha = 8$ | $\alpha = 4$ | $\alpha = 0$ | $\alpha = -4$ | $\alpha = -6$ | $\alpha = -16$ | $\alpha = -32$ |
| **Pass@1** (%) | 0.0 | 6.2 | 41.7 | 53.3 | 52.5 | 53.7 | 55.4 | 52.9 | 39.6 | 0.8 |
| **Num. Tokens** | Repetition | 1941.2 | 6232.5 | 8735.6 | 12451.2 | 14364.6 | 15144.8 | 15843.4 | 20241.7 | Repetition |

Table A4: Experimental results presented in Section 3.3. The figure version is provided in Figure 5. We compute the average performance difference of each setting compared to the original LRM in the last column (Δ Avg.)

| Methods | Performance (Pass@1 (%)↑ / Num. Tokens ↓) | | | | |
|---|---|---|---|---|---|
| | **MATH-500** | **AIME24** | **GPQA Diamond** | **LiveCodeBench** | **Δ Avg.** |
| **DeepSeek-R1-Distill-Qwen-7B** | | | | | |
| Original | 92.9 / 3403.9 | 52.5 / 12451.2 | 46.6 / 6189.3 | 48.1 / 8467.3 | - / - |
| $\alpha = -6$ | 94.0 / 4537.7 | 52.9 / 15843.4 | **49.2** / 8277.0 | **49.2** / 10531.4 | **+1.32** / +2169.5 |
| $\alpha = -4$ | **94.1** / 4193.6 | **55.4** / 15144.8 | 48.3 / 7732.3 | 49.0 / 9868.7 | **+1.69** / +1606.9 |
| $\alpha = -2$ | 93.5 / 3885.1 | 53.7 / 14364.6 | 47.5 / 6996.6 | 48.7 / 9152.4 | **+0.86** / +971.8 |
| $\alpha = 4$ | 91.4 / 2784.9 | 50.4 / 10942.7 | 47.8 / 5016.2 | 46.0 / 7152.0 | -1.12 / -1154.0 |
| $\alpha = 8$ | 89.4 / 2232.9 | 53.3 / 8735.6 | 44.8 / 3712.3 | 43.5 / 5857.3 | -2.27 / -2493.4 |
| $\alpha = 12$ | 85.5 / 1803.2 | 42.1 / 7824.8 | 44.4 / 2425.0 | 40.6 / 4522.2 | -6.87 / -3484.1 |
| $\alpha = 16$ | 81.1 / 1410.5 | 41.7 / 6232.5 | 38.0 / 1462.1 | 34.4 / 3005.5 | -11.2 / -4600.3 |
| 1xWait | 92.7 / 3744.2 | 52.1 / 12704.0 | 45.9 / 9376.0 | 48.1 / 8726.6 | -0.33 / +1009.8 |
| 2xWait | 92.7 / 5066.2 | 51.2 / 14340.9 | 45.6 / 13847.1 | 48.1 / 9029.7 | -0.63 / +2943.1 |
| 3xWait | 92.5 / 7436.3 | 50.8 / 16372.6 | 43.9 / 18294.7 | 48.0 / 9547.3 | -1.23 / +5284.8 |
| **DeepSeek-R1-Distill-Qwen-32B** | | | | | |
| Original | 94.2 / 2947.3 | 69.2 / 10679.6 | 61.1 / 5493.1 | 72.4 / 6897.1 | - / - |
| $\alpha = -6$ | **95.9** / 5020.2 | 71.7 / 15342.5 | **62.3** / 8501.8 | 72.6 / 9153.0 | **+1.40** / +3000.1 |
| $\alpha = -4$ | 95.8 / 4199.4 | **73.8** / 14254.7 | 62.2 / 7500.3 | **73.0** / 8477.1 | **+1.98** / +2103.6 |
| $\alpha = -2$ | 94.9 / 3533.5 | 68.2 / 11984.2 | 60.7 / 6428.3 | 72.6 / 7615.4 | -0.10 / +1270.5 |
| $\alpha = 4$ | 92.0 / 2088.7 | 60.8 / 8000.1 | 61.2 / 3793.4 | 69.8 / 5412.6 | -3.27 / -1680.6 |
| $\alpha = 8$ | 87.6 / 1495.8 | 56.7 / 5950.1 | 58.8 / 2126.4 | 67.2 / 4125.1 | -6.65 / -3079.9 |
| $\alpha = 12$ | 81.4 / 1034.5 | 38.3 / 3893.7 | 53.2 / 1237.5 | 61.9 / 2830.2 | -15.5 / -4255.3 |
| $\alpha = 16$ | 74.3 / 720.5 | 21.7 / 1784.1 | 46.6 / 786.1 | 54.7 / 1448.9 | -24.9 / -5319.4 |
| 1xWait | 94.4 / 3634.4 | 68.3 / 11323.3 | 61.5 / 6671.6 | 72.6 / 7080.4 | -0.02 / +673.2 |
| 2xWait | 94.9 / 8457.8 | 69.2 / 14453.3 | 59.5 / 9529.3 | 72.6 / 7372.3 | -0.18 / +3448.9 |
| 3xWait | 95.0 / 15282.1 | 69.2 / 20468.7 | 57.6 / 14057.9 | 72.5 / 8417.4 | -0.65 / +8052.3 |
| **QwQ-32B** | | | | | |
| Original | 97.4 / 4305.0 | 76.7 / 13627.3 | 62.7 / 7968.7 | 89.2 / 7190.2 | - / - |
| $\alpha = -6$ | **97.7** / 5000.8 | 78.3 / 15861.4 | 63.4 / 9044.0 | **90.1** / 8149.0 | **+0.89** / +1241.0 |
| $\alpha = -4$ | 97.4 / 4704.1 | **78.7** / 15191.7 | **64.0** / 8707.9 | 90.0 / 7919.0 | **+1.05** / +857.9 |
| $\alpha = -2$ | 97.4 / 4438.3 | **78.7** / 13896.0 | 62.9 / 8318.9 | 89.8 / 7512.7 | **+0.72** / +268.7 |
| $\alpha = 4$ | 97.2 / 3921.4 | 76.2 / 12191.6 | 63.3 / 7083.8 | 88.6 / 6382.6 | -0.17 / -878.0 |
| $\alpha = 8$ | 97.3 / 3715.3 | 73.3 / 12068.0 | 62.6 / 6258.9 | 87.8 / 5870.6 | -1.25 / -1294.6 |
| $\alpha = 12$ | 96.9 / 3569.0 | 74.2 / 11530.1 | 61.5 / 5629.7 | 86.4 / 5523.5 | -1.75 / -1709.7 |
| $\alpha = 16$ | 96.6 / 3444.1 | 71.2 / 11299.9 | 60.7 / 5220.5 | 83.9 / 5277.1 | -3.40 / -1962.4 |
| 1xWait | 97.3 / 4540.5 | 76.2 / 13868.8 | 63.8 / 8369.2 | 89.2 / 7501.0 | **+0.13** / +297.1 |
| 2xWait | 97.3 / 5333.3 | 75.8 / 14402.5 | 63.2 / 8824.2 | 89.2 / 7789.8 | -0.13 / +814.7 |
| 3xWait | 97.3 / 7604.0 | 75.8 / 16450.0 | 62.8 / 9705.2 | 89.0 / 8093.7 | -0.28 / +2190.4 |

# B Supplementary materials for Section 4

## B.1 Details on sliding-window based adaptive control algorithm

---

**Algorithm 1** Sliding-window based Adaptive Control Algorithm

---

**Require:**

    $N$ transformer layers $\{F^i(\cdot)\}_{i=1}^N$, embedding layer $E(\cdot)$, vocabulary head $\phi(\cdot)$

    End-of-sequence token EOS, user prompt $p$, maximum length max_len

    Layers to control $L^c \subseteq \{1, \ldots, N\}$, early layers $L^e \subseteq \{1, \ldots, N-1\}$

    Reading vectors $\{v^l \mid l \in L^c\}$

    Steering intensities: initial $\alpha_s$, min $\alpha_{\min}$, max $\alpha_{\max}$

    Step size $s$, window size $W$, outlier detection threshold $\lambda$

**Initialize:**

  1: $\mathbf{x} \leftarrow p$                                                 ▷ Input sequence

  2: $\mathbf{y} \leftarrow \emptyset$                                                ▷ Output sequence

  3: $\alpha \leftarrow \alpha_s$                                         ▷ Current steering intensity

  4: window $\leftarrow \emptyset$                           ▷ Sliding window of difficulty scores

  5: **while** length($\mathbf{x}$) < max_len **do**

  6:     $h_{<t}^0 \leftarrow E(\mathbf{x})$                          ▷ Get embeddings

  7:     $H^e \leftarrow \emptyset$                       ▷ Initialize early layer states

  8:     **for** $i \leftarrow 1$ **to** $N$ **do**             ▷ Process through all layers

  9:         $h_t^i \leftarrow F^i(h_t^{i-1}, h_{<t}^{i-1})$          ▷ Standard forward pass

10:         **if** $i \in L^c$ **then**

11:             $h_t^i \leftarrow h_t^i + \alpha \cdot v^i$        ▷ Apply representation control

12:         **end if**

13:         **if** $i \in L^e$ **then**

14:             $H^e \leftarrow H^e \cup \{h_t^i\}$        ▷ Collect early layer states

15:         **end if**

16:     **end for**

17:     JS_Div $\leftarrow$ ComputeJSDivergence($H^e, h_t^N$)      ▷ Using Equation (3)

18:     $d(x_t) \leftarrow$ ComputeTokenDifficulty(JS_Div)      ▷ Using Equation (4)

19:     **if** $d(x_t) > \text{mean(window)} + \lambda \cdot \text{std(window)}$ **then**

20:         $\alpha \leftarrow \alpha_s$            ▷ Slow down for careful reasoning

21:     **else**

22:         $\alpha \leftarrow \min(\alpha_{\max}, \alpha + s)$      ▷ Speed up for efficient reasoning

23:     **end if**

24:     Update window with $d(x_t)$, maintaining size $W$

25:     $x_t \leftarrow \phi(h_t^N)$                 ▷ Generate next token

26:     $\mathbf{x} \leftarrow \mathbf{x} \cup \{x_t\}$              ▷ Append to input

27:     $\mathbf{y} \leftarrow \mathbf{y} \cup \{x_t\}$

28:     **if** $x_t = $ EOS **then**

29:         **break**

30:     **end if**

31: **end while**

32: **return** $\mathbf{y}$

---

We provide a detailed pseudocode of our sliding-window-based adaptive control algorithm in Algorithm 1. During generation, we keep track of the reasoning difficulties of the most recent $k$ tokens in the sliding window. In each forward pass, we first steer the representations from the target control layers $L^c$ using the current intensity $\alpha$ (lines 10-12), and then we collect the logit projections from the early layers $L^e$ (lines 13-15). After computing the current token reasoning difficulty $d(x_t)$ using Equation (4) (lines 17-18), we compare it to the current sliding window threshold (line 19). If $d(x_t)$ is larger than the threshold, we set the next steering intensity to the lower bound $\alpha_{\min}$ (similar to stepping on the brake). Otherwise, we increase the thinking speed by a predefined increment size $s$ (lines 19-23).

The full hyperparameter settings for the above algorithm are shown in Table B1.

Table B1: Parameter settings for the adaptive control algorithm.

| Parameters | Values | | | |
| --- | --- | --- | --- | --- |
| | DeepSeek-R1-Distill-Qwen-7B | Qwen3-8B | DeepSeek-R1-Distill-Qwen-32B | QwQ-32B |
| Layers to control $L^c$ | $[19, 27]$ | $[26, 35]$ | $[44, 63]$ | $[49, 63]$ |
| Early layers $L^e$ | $[1, 14]$ | $[1, 18]$ | $[22, 42]$ | $[22, 42]$ |
| Initial steering intensity $\alpha_s$ | | | -4 | |
| Min. steering intensity $\alpha_{\min}$ | | | -4 | |
| Max. steering intensity $\alpha_{\max}$ | | | 4 | |
| Adjustment step size $s$ | | | 2 | |
| Window size $W$ | | | 8 | |
| Outlier detection threshold $\lambda$ | 2.0 | 2.0 | 1.5 | 1.5 |

## B.2 Ablation study on early layer bucket selection

Table B2: Ablation results of early layer bucket selection. Performances that surpass the original LRMs' are highlighted in **bold**. We outline the actual parameter settings we use in Table 2 by ▢.

| Methods | AMC Performance (Pass@1 (%)↑ / Num. Tokens ↓) |
| --- | --- |
| **DeepSeek-R1-Distill-Qwen-7B** | 81.9 / 7554.8 |
| *w.* **Adaptive control** with early layers $[1, \ 14]$ | **82.8** / 6414.2 |
| *w.* **Adaptive control** with early layers $[14, 27]$ | **82.2** / 6775.3 |
| **Qwen3-8B** | 89.5 / 10363.7 |
| *w.* **Adaptive control** with early layers $[1, \ 18]$ | **89.8** / 9862.6 |
| *w.* **Adaptive control** with early layers $[18, 35]$ | **89.6** / 9993.7 |
| **DeepSeek-R1-Distill-Qwen-32B** | 88.1 / 6706.4 |
| *w.* **Adaptive control** with early layers $[1, \ 21]$ | 87.0 / 5516.3 |
| *w.* **Adaptive control** with early layers $[22, 42]$ | **88.8** / 5734.4 |
| *w.* **Adaptive control** with early layers $[43, 64]$ | 87.1 / 5994.7 |
| **QwQ-32B** | 94.4 / 8735.6 |
| *w.* **Adaptive control** with early layers $[1, \ 21]$ | 92.9 / 8519.1 |
| *w.* **Adaptive control** with early layers $[22, 42]$ | **94.4** / 8413.1 |
| *w.* **Adaptive control** with early layers $[43, 64]$ | **94.4** / 8420.8 |

To determine the range of early layers for logits contrasting, we group transformer layers into buckets and choose the optimal layer bucket based on validation results on the AMC problem set [17]. For DeepSeek-R1-Distill-Qwen-7B (28 layers), we set 2 buckets: $[1, 14]$ and $[14, 27]$. For Qwen3-8B (36 layers), we set 2 buckets: $[1, 18]$ and $[18, 35]$. For DeepSeek-R1-Distill-Qwen-32B and QwQ-32B (64 layers), we set 3 buckets: $[1, 21]$, $[22, 42]$, and $[43, 63]$. For efficiency, we sample logits from every other layer within each bucket. The validation results are shown in Table B2. Based on these results, we decide to use the layer bucket $[1, 14]$ for DeepSeek-R1-Distill-Qwen-7B, $[1, 18]$ for Qwen3-8B, and $[22, 42]$ for DeepSeek-R1-Distill-Qwen-32B and QwQ-32B.

## B.3 Ablation study on the effect of reasoning difficulty-based adjustments

To demonstrate the effect of our reasoning difficulty measurement in our control algorithm, we compare it with a random adjustment baseline. The random baseline sets the steering intensity $\alpha_t$ at each decoding step to $\alpha_{\min}$ or $\alpha_{\max}$ based on coin flips. Other parameters remain the same as

Table B3: Comparison of reasoning difficulty-based control method with random control baseline on DeepSeek-R1-Distill-Qwen-7B. Performances that surpass the original LRMs' are highlighted in **bold**. We outline the actual parameter settings we use in Table 2 by ▨.

| Methods | Performance (Pass@1 (%)↑ / Num. Tokens ↓) | | |
| --- | --- | --- | --- |
| | MATH-500 | AIME24 | GPQA Diamond |
| **DeepSeek-R1-Distill-Qwen-7B** | 92.9 / 3403.9 | 52.5 / 12451.2 | 46.6 / 6189.3 |
| *w.* Adaptive control | **93.7 / 3122.8** | **53.8 / 10850.9** | **48.8 / 5421.6** |
| *w.* Random control | **93.4** / 3862.6 | 52.5 / 14505.1 | **47.5** / 7113.2 |

in Table B1. The results are shown in Table B3. The reasoning difficulty-based method consistently outperforms the random baseline, validating the effectiveness of our reasoning difficulty measurement.

## B.4 Ablation study on the outlier detection threshold $\lambda$

Table B4: Ablation results of outlier detection threshold $\lambda$. Performances that surpass the original LRMs' are highlighted in **bold**. We outline the actual parameter settings we use in Table 2 by ▨.

| Methods | Performance (Pass@1 (%)↑ / Num. Tokens ↓) | |
| --- | --- | --- |
| | MATH-500 | AIME24 |
| **DeepSeek-R1-Distill-Qwen-7B** | 92.9 / 3403.9 | 52.5 / 12451.2 |
| *w.* Adaptive control ($\lambda = 1.0$) | **93.5** / 3498.6 | **52.6** / 12659.6 |
| *w.* Adaptive control ($\lambda = 2.0$) | **93.7** / 3122.8 | **53.8** / 10850.9 |
| *w.* Adaptive control ($\lambda = 3.0$) | **92.9** / 2927.7 | 51.3 / 11132.6 |
| **DeepSeek-R1-Distill-Qwen-32B** | 94.2 / 2947.3 | 69.2 / 10679.6 |
| *w.* Adaptive control ($\lambda = 1.0$) | **94.8** / 2938.0 | **69.4** / 9883.6 |
| *w.* Adaptive control ($\lambda = 1.5$) | **94.9** / 2732.9 | **70.7** / 9322.5 |
| *w.* Adaptive control ($\lambda = 2.0$) | 94.1 / 2637.3 | 64.4 / 8972.9 |
| *w.* Adaptive control ($\lambda = 3.0$) | **94.3** / 2478.4 | 66.1 / 7848.0 |

We show the results of our adaptive control method under different outlier detection thresholds $\lambda$ in Table B4. We observe a clear trend: as $\lambda$ decreases, our algorithm identifies more tokens as difficult, leading to three correlated effects: (1) reduced overall reasoning speed, (2) increased response length, and (3) generally improved accuracy. These results demonstrate how threshold selection directly mediates the speed-accuracy tradeoff in our framework.

## B.5 Ablation study on the maximum steering intensity $\alpha_{\text{max}}$

Our ablation study on maximum steering intensity $\alpha_{\text{max}}$ is presented in Table B5. The results demonstrate two key trends: (1) response lengths decrease with increasing $\alpha_{\text{max}}$, indicating accelerated overall thinking speed, and (2) accuracy improves when constraining $\alpha_{\text{max}}$, confirming the benefits of deeper, slower thinking. These findings collectively validate the effectiveness of our adaptive control approach.

## B.6 Case studies on the adaptive control algorithm

To provide a more straightforward understanding of the effect of our adaptive control algorithm, we present the following case studies to analyze the impact of our dynamic adjustment. For each case, we first apply our adaptive control algorithm to the DeepSeek-R1-Distill-Qwen-7B model on the AIME24 benchmark. We then select the top 10 tokens with the strongest reasoning difficulty signals. Among those, we identify the tokens we believe to be the **"turning points"** that are keys to the

Table B5: Ablation results of maximum steering intensity $\alpha_{\mathrm{max}}$. Performances that surpass the original LRMs' are highlighted in **bold**. We outline the actual parameter settings we use in Table 2 by ▨.

| Methods | Performance (Pass@1 (%)↑ / Num. Tokens ↓) | |
| --- | --- | --- |
| | **MATH-500** | **AIME24** |
| **DeepSeek-R1-Distill-Qwen-7B** | 92.9 / 3403.9 | 52.5 / 12451.2 |
| *w.* Adaptive control ($\alpha_{\mathrm{max}} = 0.0$) | **93.8** / 3598.0 | **54.2** / 13592.7 |
| *w.* Adaptive control ($\alpha_{\mathrm{max}} = 4.0$) | **93.7** / 3122.8 | **53.8** / 10850.9 |
| *w.* Adaptive control ($\alpha_{\mathrm{max}} = 8.0$) | **92.9** / 2800.2 | 48.3 / 10361.9 |
| *w.* Adaptive control ($\alpha_{\mathrm{max}} = 12.0$) | 92.1 / 2580.0 | 47.5 / 9097.2 |
| **DeepSeek-R1-Distill-Qwen-32B** | 94.2 / 2947.3 | 69.2 / 10679.6 |
| *w.* Adaptive control ($\alpha_{\mathrm{max}} = 0.0$) | **94.8** / 3272.2 | **69.7** / 11065.6 |
| *w.* Adaptive control ($\alpha_{\mathrm{max}} = 4.0$) | **94.9** / 2732.9 | **70.7** / 9322.5 |
| *w.* Adaptive control ($\alpha_{\mathrm{max}} = 8.0$) | **94.2** / 2306.4 | 65.8 / 8081.9 |
| *w.* Adaptive control ($\alpha_{\mathrm{max}} = 12.0$) | 93.4 / 2088.5 | 60.7 / 7608.6 |

correct answers in the solution processes. We compare the original response produced under adaptive adjustment (*i.e.*, deceleration at these tokens) with the response produced without adjustment (*i.e.*, maintaining acceleration ($\alpha = 4$)). As shown in Figure B1, B2, and B3, the deceleration adjustments made by our control algorithm effectively stimulate the LRM's analytical and reflective capabilities, steering the reasoning process toward correct solutions. These strategic pauses in the reasoning flow optimally utilize the LRM's reasoning abilities, thereby preventing incorrect answers or token waste in erroneous solution exploration. We believe these characteristics are the key elements of our adaptive control method in achieving the ideal accuracy-efficiency trade-off.

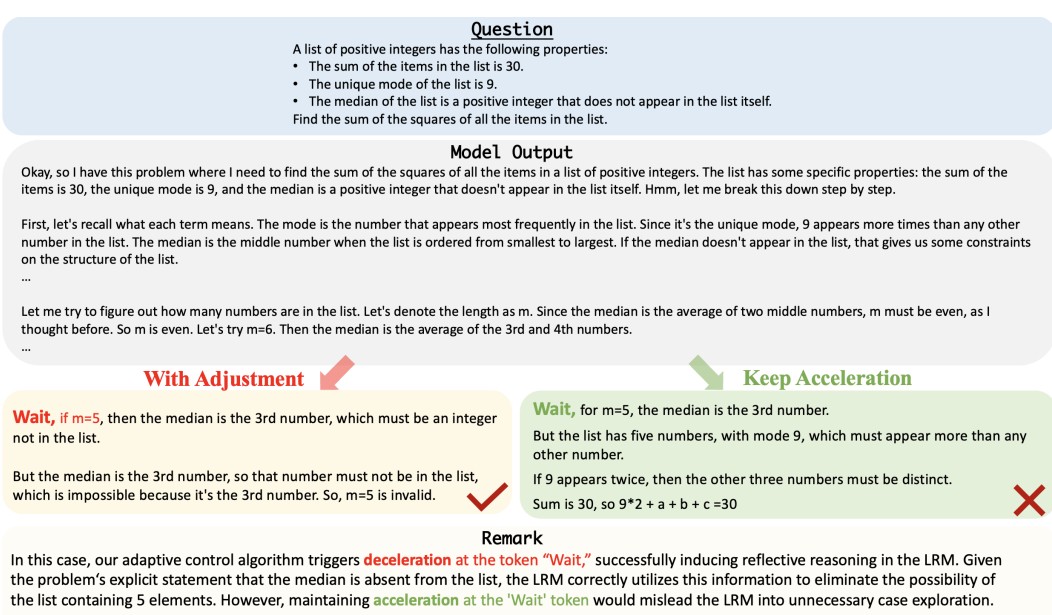

Figure B1: Case study 1 on the effectiveness of our dynamic adjustments to LRM reasoning. The token highlighted in **red and bold** is identified as having high reasoning difficulty, prompting our deceleration adjustment in the subsequent tokens. We compare the results produced by our adaptive control algorithm with those produced by maintaining acceleration.

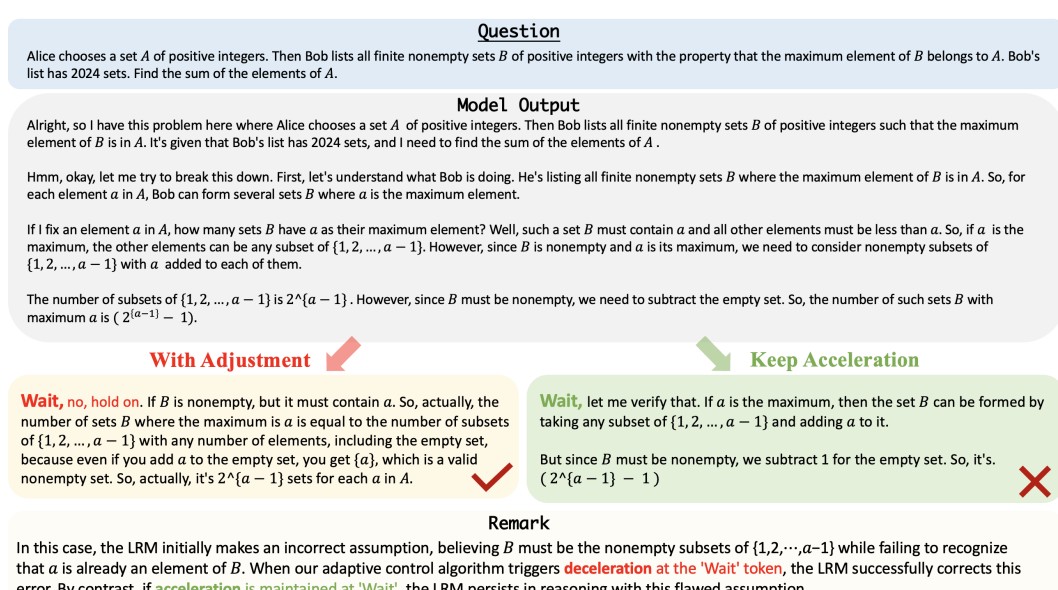

Figure B2: Case study 2 on the effectiveness of our dynamic adjustments to LRM reasoning.

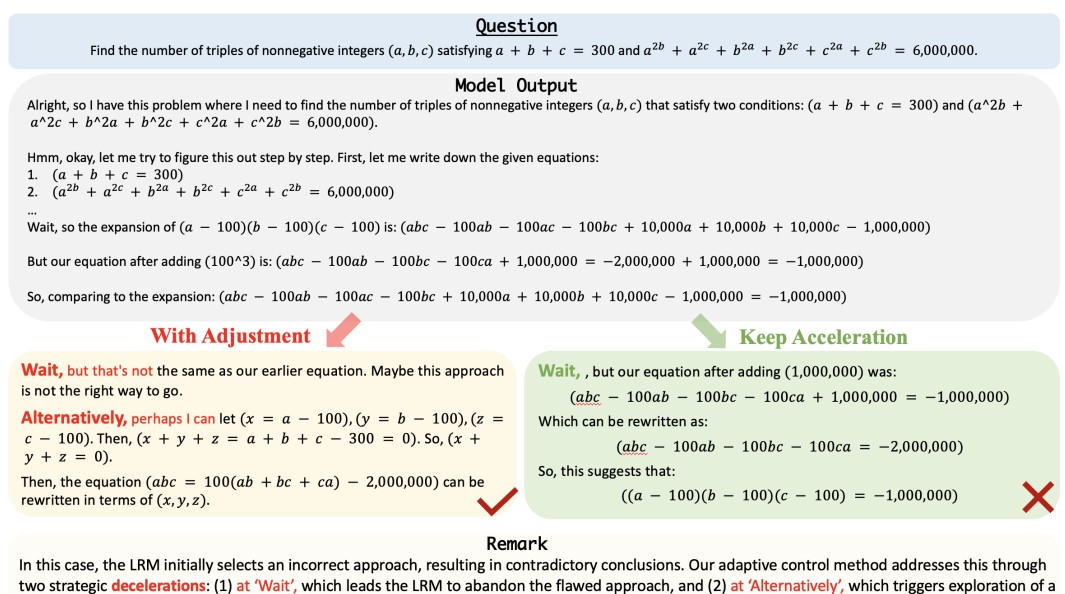

Figure B3: Case study 3 on the effectiveness of our dynamic adjustments to LRM reasoning.

## B.7 Analysis of switching dynamics in the adaptive control algorithm

To quantify how our adaptive control algorithm influences the reasoning flow of the underlying model, we analyze its transitions between fast- and slow-thinking modes during inference. We use **DeepSeek-R1-Distill-Qwen-7B** throughout the experiments.

**Switching frequencies**    We first analyze the model's mode switch frequency on math benchmarks, finding averages of **578.6** (AIME24) and **167.2** (MATH-500) switches per question. The higher frequency for AIME24 reflects its greater complexity, requiring more slow-thinking engagement. Next, to better understand the transition dynamics, we examine how switching frequency varies throughout the thought process. We divide the responses into three temporal segments: *start* (first

25%), *middle* (25%-75%), and *end* (last 25%). We then calculate the average token intervals between these switches (with lower values indicating more frequent switches). The results are presented in Table B6. We found that (1) the model most frequently enters slow-thinking mode at the beginning, likely due to initial problem analysis requiring deeper processing; (2) the middle segment also shows high switching frequency, reflecting active reasoning during problem-solving; and (3) switching drops in the final segment, as the model converges on a solution and generates fewer new thoughts.

Table B6: Transition dynamics.    Table B7: Frequent slow-thinking trigger tokens.

| Dataset | Avg. # of tokens between switches | | | Type | Most frequent words |
|---------|-------|------|------|------|------|
| | Start | Mid | End | Calculation | sqrt, denominator, $\approx$, triangle, ... |
| **AIME24** | 17.8 | 20.9 | 54.9 | Analysis | Problem, seems, because, find, ... |
| **MATH-500** | 14.5 | 22.5 | 41.7 | Reflection | Wait, Alternatively, no, maybe ... |

**Relationship between mode transitions and model outputs**    To analyze the relationship between internal thinking modes and model outputs, we first identify the top-100 tokens that most frequently trigger slow-thinking transitions in Table B7. Our results demonstrate that models tend to switch to slow-thinking for (1) mathematical computations, (2) logical deductions, and (3) triggering certain reflection behaviors. To understand how this control signal influences the generation, we identify the top-5 most frequent tokens immediately after slow-thinking switches:

```
[" no", " maybe", " let", " perhaps", " but"]
```

These hesitation markers consistently signal reflections and reconsiderations in models' CoTs.

To assess the alignment between internal mode switches and external outputs, we measured how often the token "Wait", commonly used as a marker of uncertainty [4, 19], coincides with actual transitions to slow-thinking mode. Surprisingly, on AIME24, the model outputs "Wait" 55.3 times per question on average, but only **1 in 12.2** instances aligns with a true mode switch. On MATH-500, the ratio is **1 in 15.0**. These results suggest that (1) the overuse of "Wait" reflects the overthinking behavior [4], and (2) analyses relying on output tokens to infer reasoning states [4, 38] may be misleading, given the weak correlation with internal cognitive transitions. We hope these findings will provide additional insights for interpreting LLMs' reasoning behaviors and inspire future research in this area.

## C Combining thinking speed control with parallel search

To further demonstrate the effectiveness and extensibility of our thinking speed control method, we explore one practical application by integrating it with parallel search [20, 29]. This integration yields non-trivial performance improvements over standard baselines.

**Experimental settings**  We conduct experiments using **DeepSeek-R1-Distill-Qwen-32B** on both **AIME24** and **GPQA Diamond**. For each question, we sample up to $N = 64$ responses to enable parallel search. We report Pass@$k$ [3], which measures the probability of obtaining at least one correct solution among $k$ randomly selected completions. To assess the benefit of our control method under a breadth-first search regime, we set the steering intensity $\alpha = 8/12/16$ to accelerate the search process and compare against the following baselines:

- **Vanilla Generation**: Default generation from the base model.
- **NoThinking** [20]: A concurrent method that improves efficiency by skipping explicit reasoning. Specifically, it pre-fills the thought process with:

> $<$think$>$\nOkay, I think I have finished thinking.\n$<$/think$>$.

All methods use identical vLLM generation configurations as described in Section 3.3. To ensure a fair comparison while keeping the computational costs practical in daily usage, we set the maximum generation length per search to $1000/2000/3000/4000$ tokens.

**Results and Analysis**  The results are presented in Figure C4, where the y-axis denotes Pass@$k$ and the x-axis represents the average response length. We observe that the LRM's default generation performs the worst under low token budget settings, while NoThinking achieves slightly better efficiency. However, both baselines are largely surpassed by our acceleration methods across nearly all token budgets and search breadths. These results further demonstrate the superiority of our approach in achieving higher efficiency and greater extensibility, highlighting its potential to support extreme test-time scaling.

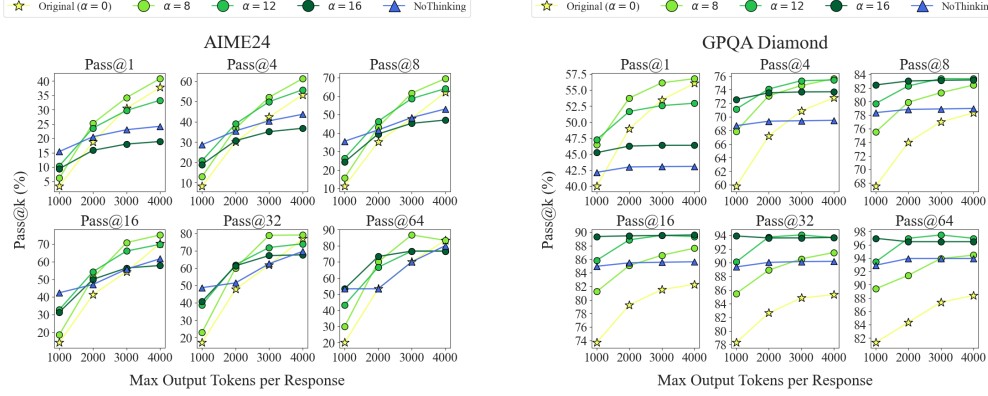

Figure C4: Parallel search results with thinking speed control. The plots show the relationship between response length (x-axis, average token count) and reasoning performance (y-axis, Pass@$k$) under different search space sizes. Key annotations: (1)"★": Baseline model performance. (2)"▲": Performance of NoThinking [20]. (3)"●∼●": Representation control with different steering intensity $\alpha$. Our control method consistently outperforms the baselines across different search settings.

# D   Example generations

## D.1   Different types of CoT

Below, we present a question from the MATH training set and three different responses sampled from (1) Qwen2.5-MATH-7B-Instruct [42], and (2) slow response (*i.e.*, regular response) from DeepSeek-R1-Distill-Qwen-7B [7] and (3) fast response from DeepSeek-R1-Distill-Qwen-7B.

**Question:**

> What is the greatest integer $x$ such that $|6x^2 - 47x + 15|$ is prime?

**Slow response from DeepSeek-R1-Distill-Qwen-7B:**

> Okay, so I have this problem here: I need to find the greatest integer $x$
> such that the absolute value of $6x^2 - 47x + 15$ is a prime number. Hmm, prime
> numbers are numbers greater than 1 that have no positive divisors other than
> 1 and themselves. So, I need to find $x$ such that $|6x^2 - 47x + 15|$ is prime.
> And specifically, I'm looking for the greatest such integer $x$.
> First, let me consider the quadratic expression $6x^2 - 47x + 15$. Since it's
> a quadratic, it can be factored or maybe I can find its roots to understand
> its behavior. Maybe factoring will help me see if the quadratic can be
> broken down into simpler terms, which might make it easier to analyze.
> Let me try factoring $6x^2 - 47x + 15$. To factor a quadratic of the form $ax^2 +$
> $bx + c$, I look for two numbers that multiply to $a \times c$ and add up to $b$. Here,
> $a = 6$, $b = -47$, and $c = 15$. So, $a \times c = 6 \times 15 = 90$. I need two numbers that
> multiply to 90 and add up to -47.
> Let me think: 90 is positive, so either both numbers are negative or both
> positive. But since they add up to -47, which is negative, they must both
> be negative. So, looking for negative factors of 90 that add up to -47.
> Factors of 90: 1 and 90, 2 and 45, 3 and 30, 5 and 18, 6 and 15, 9 and 10.
> Let me try pairs:
> - (-1, -90): Sum is -91. Not -47. - (-2, -45): Sum is -47. Oh, wait,
> that's exactly what I need!
> So, I can rewrite the middle term using these numbers. So, $-47x$ can be
> written as $-2x - 45x$. Therefore, the quadratic becomes:
> $6x^2 - 2x - 45x + 15$
> Now, let's group the terms:
> $(6x^2 - 2x) + (-45x + 15)$
> Factor out the greatest common factor (GCF) from each group:
> From the first group, $6x^2 - 2x$, the GCF is $2x$:
> $2x(3x - 1)$
> From the second group, $-45x + 15$, the GCF is -15:
> $-15(3x - 1)$
> So, putting it all together:
> $2x(3x - 1) - 15(3x - 1)$
> Now, factor out the common binomial factor $(3x - 1)$:
> $(3x - 1)(2x - 15)$
> So, the quadratic factors as $(3x - 1)(2x - 15)$. Therefore, $6x^2 - 47x + 15 = (3x -$
> $1)(2x - 15)$.
> Since the problem is about the absolute value of this expression being prime,
> let's denote $N = |(3x - 1)(2x - 15)|$. We need $N$ to be a prime number.
> Prime numbers have only two positive divisors: 1 and themselves. So, for
> $N$ to be prime, the product $(3x - 1)(2x - 15)$ must be either a prime number or
> its negative counterpart, since we're taking the absolute value. Therefore,
> the product must be either $p$ or $-p$, where $p$ is a prime number.
> Given that $(3x - 1)(2x - 15)$ is a product of two linear terms in $x$, for the
> product to be prime, one of the factors must be $\pm 1$ and the other must be $\pm p$.
> Because primes have only two distinct positive divisors, 1 and themselves,

so the only way a product of two integers is prime is if one of them is 1 or -1, and the other is the prime or its negative.

Therefore, we can set up equations where either $3x - 1 = \pm 1$ or $2x - 15 = \pm 1$, and solve for $x$. Then, for each solution, check if the other factor is a prime number or its negative.

Let me first solve $3x - 1 = 1$:  $3x - 1 = 1$ Add 1 to both sides:  $3x = 2$ Divide by 3:  $x = \frac{2}{3}$ But $x$ needs to be an integer, so this solution is invalid.

Next, solve $3x - 1 = -1$:  $3x - 1 = -1$ Add 1 to both sides:  $3x = 0$ Divide by 3: $x = 0$ Check the other factor:  $2x - 15 = 2(0) - 15 = -15$ So, $(3x - 1)(2x - 15) = (-1)(-15) = 15$ But 15 is not a prime number, so this is invalid.

Now, solve $2x - 15 = 1$:  $2x - 15 = 1$ Add 15 to both sides:  $2x = 16$ Divide by 2: $x = 8$ Check the other factor:  $3x - 1 = 3(8) - 1 = 24 - 1 = 23$ So, $(3x-1)(2x-15) = 23 \times 1 = 23$ 23 is a prime number, so this is valid.  So, $x = 8$ is a candidate.

Next, solve $2x - 15 = -1$:  $2x - 15 = -1$ Add 15 to both sides:  $2x = 14$ Divide by 2:  $x = 7$ Check the other factor:  $3x - 1 = 3(7) - 1 = 21 - 1 = 20$ So, $(3x - 1)(2x - 15) = 20 \times (-1) = -20$ Absolute value is 20, which is not prime.

So, from the four cases, only $x = 8$ gives a prime number.  But wait, before I conclude, maybe I should consider other possibilities where one of the factors is a prime and the other is -1, but I think I've covered all cases where each factor is either 1 or -1.

But just to be thorough, let me analyze if there could be other integer values of $x$ where the product $(3x - 1)(2x - 15)$ is prime, without necessarily one of the factors being 1 or -1.

Suppose both $3x - 1$ and $2x - 15$ are primes or their negatives, but their product is also prime.  Well, the only way two integers multiply to a prime is if one is 1 or -1 and the other is the prime or its negative.  So, that covers all cases.

Therefore, the only integer $x$ that satisfies the condition is $x = 8$.  But the problem asks for the greatest integer $x$, so maybe 8 is the answer.  But wait, let me check higher integers beyond 8 to see if any of them might give a prime.

Let me test $x = 9$:  Compute $6(9)^2 - 47(9) + 15$ First, 6*81 = 486 47*9 = 423 So, 486 - 423 + 15 = (486 - 423) + 15 = 63 + 15 = 78 |78| = 78, which is not prime.

x = 10:  6*100 = 600 47*10 = 470 600 - 470 + 15 = 130 + 15 = 145 |145| = 145, which is not prime.

x = 11:  6*121 = 726 47*11 = 517 726 - 517 + 15 = 209 + 15 = 224 |224| = 224, not prime.

x = 12:  6*144 = 864 47*12 = 564 864 - 564 + 15 = 300 + 15 = 315 Not prime.

x = 13:  6*169 = 1014 47*13 = 611 1014 - 611 + 15 = 403 + 15 = 418 Not prime.

x = 7, we already saw it gives 20 in the other factor, which is not prime.

x = 6:  6*36 = 216 47*6 = 282 216 - 282 + 15 = (-66) + 15 = -51 | -51 | = 51, not prime.

x = 5:  6*25 = 150 47*5 = 235 150 - 235 + 15 = (-85) + 15 = -70 | -70 | = 70, not prime.

x = 4:  6*16 = 96 47*4 = 188 96 - 188 + 15 = (-92) + 15 = -77 | -77 | = 77, not prime.

x = 3:  6*9 = 54 47*3 = 141 54 - 141 + 15 = (-87) + 15 = -72 | -72 | = 72, not prime.

x = 2:  6*4 = 24 47*2 = 94 24 - 94 + 15 = (-70) + 15 = -55 | -55 | = 55, not prime.

x = 1:  6*1 = 6 47*1 = 47 6 - 47 + 15 = (-41) + 15 = -26 | -26 | = 26, not prime.

x = 0:  6*0 = 0 47*0 = 0 0 - 0 + 15 = 15 |15| = 15, not prime.

x = -1:  6*(-1)$\hat{2}$ = 6*1 = 6 47*(-1) = -47 6 - (-47) + 15 = 6 + 47 + 15 = 68 |68| = 68, not prime.

x = -2:  6*4 = 24 47*(-2) = -94 24 - (-94) + 15 = 24 + 94 + 15 = 133 |133| = 133, which is...  wait, is 133 a prime?

Wait, 133 divided by 7 is 19, because 7*19 = 133.  So, 133 is not prime.

x = -3:  6*9 = 54 47*(-3) = -141 54 - (-141) + 15 = 54 + 141 + 15 = 210 |210| = 210, not prime.

```
x = -4:  6*16 = 96 47*(-4) = -188 96 - (-188) + 15 = 96 + 188 + 15 = 300 - 1
= 299?  Wait, 96 + 188 is 284, plus 15 is 299.
Is 299 prime?  Let's check:  299 divided by 13 is 23, because 13*23 = 299.
So, 299 is composite.
x = -5:  6*25 = 150 47*(-5) = -235 150 - (-235) + 15 = 150 + 235 + 15 = 400
|400| = 400, not prime.
```
Hmm, so all integer values of $x$ less than 8 either result in composite numbers or negative numbers whose absolute values are composite. So, the only integer $x$ that gives a prime is 8, as found earlier.

Wait, but hold on. Let me consider if $x$ could be such that one of the factors is -1, leading to the other factor being a negative prime. But we already considered that when we set $2x - 15 = -1$, which gave $x = 7$, but that resulted in the other factor being 20, which is composite. So, even in that case, it didn't work.

Alternatively, perhaps I should also check if the quadratic could be equal to a prime number without one of the factors being 1 or -1. But as I thought earlier, since the quadratic factors into two linear terms, their product can only be prime if one is 1 or -1. So, I think that covers all possibilities.

Just to make sure I haven't missed anything, let me think about the quadratic expression again. Since $6x^2 - 47x + 15$ factors into $(3x - 1)(2x - 15)$, and for the absolute value to be prime, one of these factors must be 1 or -1. Since we've already found that $x = 8$ gives 23, which is prime, and all other integer $x$ values either give composite numbers or non-prime absolute values, I think 8 is indeed the greatest integer $x$ that satisfies the condition.

Therefore, my conclusion is that the greatest integer $x$ is 8.

**Final Answer** The greatest integer $x$ is $\boxed{8}$.

**Fast response from DeepSeek-R1-Distill-Qwen-7B:**

To determine the greatest integer $x$ such that $|6x^2 - 47x + 15|$ is a prime number, I first need to analyze the quadratic expression inside the absolute value.

I'll start by factoring the quadratic equation $6x^2 - 47x + 15$. To do this, I'll look for two numbers that multiply to $6 \times 15 = 90$ and add up to -47. The numbers -45 and -2 fit these criteria because $(-45) \times (-2) = 90$ and $-45 + (-2) = -47$.

Using these numbers, I can rewrite the quadratic as:

$$6x^2 - 45x - 2x + 15$$

Next, I'll factor by grouping:

$$(6x^2 - 45x) + (-2x + 15)$$

$$3x(2x - 15) - 1(2x - 15)$$

$$(3x - 1)(2x - 15)$$

So, the expression becomes:

$$|(3x - 1)(2x - 15)|$$

For the product $(3x - 1)(2x - 15)$ to be a prime number, one of the factors must be $\pm 1$. I'll consider both possibilities.

**Case 1:  $3x - 1 = 1$**

$$3x = 2$$

$$x = \frac{2}{3}$$

Since $x$ must be an integer, this case doesn't provide a valid solution.

**Case 2:  $3x - 1 = -1$**

$$3x = 0$$

$$x = 0$$

```
Plugging x = 0 into (2x − 15) gives:
                            2(0) − 15 = −15
                    |(3(0) − 1)(−15)| = |(−1)(−15)| = 15
15 is not a prime number.
**Case 3:   2x − 15 = 1**
                              2x = 16
                               x = 8
Plugging x = 8 into (3x − 1) gives:
                             3(8) − 1 = 23
                            |(23)(1)| = 23
23 is a prime number.
**Case 4:   2x − 15 = −1**
                              2x = 14
                               x = 7
Plugging x = 7 into (3x − 1) gives:
                             3(7) − 1 = 20
                           |(20)(−1)| = 20
20 is not a prime number.
Among the integer values of x that satisfy the condition, the greatest value
is x = 8.
```

## D.2    Same question under different steering intensities

Below we present an AIME24 question with corresponding responses from DeepSeek-R1-Distill-Qwen-7B across varying steering intensities.

**Question:**

```
Every morning Aya goes for a 9-kilometer-long walk and stops at a coffee
shop afterwards.  When she walks at a constant speed of s kilometers per
hour, the walk takes her 4 hours, including t minutes spent in the coffee
shop.  When she walks s + 2 kilometers per hour, the walk takes her 2 hours
and 24 minutes, including t minutes spent in the coffee shop.  Suppose Aya
walks at s +  ½ kilometers per hour.  Find the number of minutes the walk
takes her, including the t minutes spent in the coffee shop.
```

**Original response ($\alpha = 0$):**

```
Alright, so I've got this problem about Aya going for a walk every morning.
Let me try to break it down step by step.  Sometimes these word problems can
be a bit tricky, but taking it one piece at a time should help.
First, the problem says that Aya goes for a 9-kilometer walk every morning
and then stops at a coffee shop.  When she walks at a constant speed of s
kilometers per hour, the entire trip (walking plus coffee shop) takes her 4
hours.  Out of these 4 hours, t minutes are spent in the coffee shop.  Then,
when she increases her walking speed to s + 2 kilometers per hour, the total
time becomes 2 hours and 24 minutes, still including t minutes in the coffee
shop.  The question is asking, if she walks at s + 1/2 kilometers per hour,
how many minutes does the entire trip take, including the coffee shop time?
Okay, so let's try to make sense of this.  The key here is that the time
spent in the coffee shop, t, remains constant in both scenarios.  So, the
difference in total time must come from the time she spends walking.  That
means if her walking time decreases when she walks faster, the total time
(walking + coffee shop) also decreases.
```

Let me denote some variables to make this clearer.
Let's let:  - s = walking speed in km/h (first scenario) - s + 2 = walking
speed in km/h (second scenario) - s + 0.5 = walking speed in km/h (third
scenario, which is what we need to find) - t = minutes spent in the coffee
shop - d = distance walked, which is given as 9 km in both cases
Wait, actually, the distance walked is 9 km in all cases, so that's constant.
So, the total time is 4 hours in the first case and 2 hours and 24 minutes
in the second case.  Let me convert those times to hours because the speed
is in km/h.
4 hours is straightforward, it's 4 hours.  2 hours and 24 minutes:  24
minutes is 24/60 hours, which is 0.4 hours.  So, 2.4 hours in total.
Now, the total time is the sum of walking time and coffee shop time.  So, in
the first case, the total time is 4 hours, which is equal to (walking time
at speed s) + t (coffee shop time).  Similarly, in the second case, total
time is 2.4 hours, which is equal to (walking time at speed s + 2) + t.
Since t is the same in both cases, we can set up equations.
Let me write that out.
First scenario:  Total time = walking time + t 4 hours = (9 km) / (s km/h) +
t (in hours)
Wait, hold on.  t is in minutes.  Hmm, so maybe I should convert t to hours
as well to keep the units consistent.
Let me denote t as hours.  So, t hours is the time she spends in the coffee
shop.
Therefore, in the first scenario:  4 = (9 / s) + t
In the second scenario:  2.4 = (9 / (s + 2)) + t
So, now I have two equations:
1) 4 = (9 / s) + t
2) 2.4 = (9 / (s + 2)) + t
Since both equations equal to t, I can set them equal to each other:
4 - (9 / s) = 2.4 - (9 / (s + 2))
Let me solve for s.
First, subtract 2.4 from both sides:
4 - 2.4 - (9 / s) = - (9 / (s + 2))
Simplify 4 - 2.4:  that's 1.6.
So:
1.6 - (9 / s) = - (9 / (s + 2))
Multiply both sides by -1 to make it cleaner:
-1.6 + (9 / s) = 9 / (s + 2)
So,
(9 / s) - 1.6 = 9 / (s + 2)
Let me rewrite 1.6 as 8/5 to make calculations easier.  1.6 = 8/5.
So,
(9 / s) - (8/5) = 9 / (s + 2)
Hmm, so now I have an equation with s.  Let's try to solve for s.
First, let me denote equation:
(9/s) - (8/5) = 9/(s + 2)
Let me bring all terms to one side:
(9/s) - (8/5) - 9/(s + 2) = 0
Hmm, maybe it's better to find a common denominator and combine these terms.
But this might get complicated.  Alternatively, I can cross-multiply.
Alternatively, let me let's compute 9/s - 9/(s + 2) = 8/5
Yes, that seems manageable.
So,
9/s - 9/(s + 2) = 8/5
Factor out 9:
9 [1/s - 1/(s + 2)] = 8/5
Compute the expression in the brackets:
1/s - 1/(s + 2) = [ (s + 2) - s ] / [s(s + 2)] = 2 / [s(s + 2)]
Therefore,
9 * [2 / (s(s + 2))] = 8/5
Simplify:
18 / [s(s + 2)] = 8/5
Cross-multiplying:

```
18 * 5 = 8 * s(s + 2)
Compute 18 * 5:  90
So,
90 = 8s(s + 2)
Divide both sides by 2:
45 = 4s(s + 2)
So,
4s(s + 2) = 45
Expand the left side:
```
$4s^2 + 8s = 45$
Bring all terms to one side:
$4s^2 + 8s - 45 = 0$
Now, we have a quadratic equation:   $4s^2 + 8s - 45 = 0$
Let me try to solve this quadratic equation.
Quadratic formula:

$$s = [-b\sqrt{(b^2 - 4ac)}]/(2a)$$

Here, a = 4, b = 8, c = -45
Compute discriminant:

$$b^2 - 4ac = 8^2 - 4 * 4 * (-45) = 64 + 720 = 784$$

```
sqrt(784) = 28
So,
s = [ -8 ± 28 ] / (2 * 4) = [ -8 ± 28 ] / 8
So, two solutions:
1) [ -8 + 28 ] / 8 = 20 / 8 = 2.5
2) [ -8 - 28 ] / 8 = -36 / 8 = -4.5
Since speed cannot be negative, we discard -4.5.
Therefore, s = 2.5 km/h.
So, Aya's walking speed is 2.5 km/h in the first scenario.
Now, let's find t.
From the first equation:
4 = 9 / s + t
We have s = 2.5, so:
4 = 9 / 2.5 + t
Compute 9 / 2.5:  9 divided by 2.5 is 3.6.
So,
4 = 3.6 + t
Therefore, t = 4 - 3.6 = 0.4 hours.
Convert t to minutes:  0.4 hours * 60 minutes/hour = 24 minutes.
So, t is 24 minutes.
Let me verify this with the second scenario to make sure.
Second scenario:  speed is s + 2 = 2.5 + 2 = 4.5 km/h.
Walking time:  9 / 4.5 = 2 hours.
Total time is 2 hours (walking) + 24 minutes (coffee shop) = 2 hours 24
minutes, which matches the given information.  Perfect.
So, now, the question is:  Suppose Aya walks at s + 1/2 km/h.  Find the
number of minutes the walk takes her, including the t minutes spent in the
coffee shop.
So, s is 2.5 km/h, so s + 0.5 = 3 km/h.
We need to find the total time when she walks at 3 km/h.
Again, total time = walking time + t (coffee shop time)
We know t is 24 minutes, which is 0.4 hours.
Walking time at 3 km/h:  9 / 3 = 3 hours.
Total time:  3 hours + 0.4 hours = 3.4 hours.
Convert 3.4 hours to minutes:  3 hours is 180 minutes, 0.4 hours is 24
minutes, so total is 180 + 24 = 204 minutes.
Alternatively, 3.4 hours * 60 minutes/hour = 204 minutes.
Therefore, the total time is 204 minutes.
Let me just recap to make sure I didn't make any mistakes.
1.  Converted total times to hours because speed is in km/h.  2.  Set
up two equations with t in hours.  3.  Solved for s by equating the two
expressions for t.  4.  Found s = 2.5 km/h, then t = 0.4 hours = 24 minutes.
```

5.   Calculated walking time at s + 0.5 = 3 km/h, which is 3 hours.   6.
Added coffee shop time:   3 + 0.4 = 3.4 hours = 204 minutes.
Everything seems to check out.   I think 204 minutes is the correct answer.
**Final Answer** The total time Aya's walk takes, including the coffee shop,
is $\boxed{204}$ minutes.   </think>
Aya goes for a 9-kilometer walk every morning and stops at a coffee shop.
When she walks at a constant speed of $s$ kilometers per hour, the walk takes
4 hours including $t$ minutes spent in the coffee shop.   When she walks at
$s + 2$ kilometers per hour, the walk takes 2 hours and 24 minutes, including $t$
minutes spent in the coffee shop.   We need to find the number of minutes the
walk takes her when she walks at $s + \frac{1}{2}$ kilometers per hour.
First, we convert the total times to hours:   - 4 hours is straightforward.   -
2 hours and 24 minutes is 2.4 hours.
We set up the equations for the total time:   1.   $4 = \frac{9}{s} + \frac{t}{60}$   2.   $2.4 = \frac{9}{s+2} + \frac{t}{60}$
By subtracting the second equation from the first, we get:

$$4 - 2.4 = \frac{9}{s} - \frac{9}{s+2}$$

$$1.6 = 9\left(\frac{1}{s} - \frac{1}{s+2}\right)$$

$$1.6 = 9\left(\frac{2}{s(s+2)}\right)$$

$$1.6 = \frac{18}{s(s+2)}$$

$$1.6s(s+2) = 18$$

$$4s(s+2) = 45$$

$$4s^2 + 8s - 45 = 0$$

Solving the quadratic equation:

$$s = \frac{-8 \pm \sqrt{8^2 - 4 \cdot 4 \cdot (-45)}}{2 \cdot 4}$$

$$s = \frac{-8 \pm 28}{8}$$

$$s = 2.5 \text{ km/h}$$

Finding $t$:

$$4 = \frac{9}{2.5} + \frac{t}{60}$$

$$4 = 3.6 + \frac{t}{60}$$

$$t = 0.4 \text{ hours} = 24 \text{ minutes}$$

When walking at $s + \frac{1}{2} = 3$ km/h:

$$\text{Walking time} = \frac{9}{3} = 3 \text{ hours}$$

$$\text{Total time} = 3 + 0.4 = 3.4 \text{ hours} = 3 \times 60 + 24 = 204 \text{ minutes}$$

The total time Aya's walk takes, including the coffee shop, is $\boxed{204}$ minutes.

**Response under $\alpha = 8$:**

Okay, so I have this problem about Aya's walk and coffee shop visit.   Let me
try to understand it step by step.
Every morning, Aya goes for a 9-kilometer walk and then stops at a coffee
shop.   Her walking speed affects the total time, including the time she
spends in the coffee shop.
When she walks at a constant speed of $s$ kilometers per hour, the entire
activity (walking plus coffee shop) takes 4 hours.   Out of these 4 hours, $t$

minutes are spent in the coffee shop. So, the walking time at speed $s$ is $4 - \frac{t}{60}$ hours.

Similarly, when she increases her walking speed to $s + 2$ kilometers per hour, the total time becomes 2 hours and 24 minutes. Converting 2 hours and 24 minutes to hours, that's $2 + \frac{24}{60} = 2.4$ hours. So, the walking time at speed $s + 2$ is $2.4 - \frac{t}{60}$ hours.

Our goal is to find the total time when she walks at $s + \frac{1}{2}$ kilometers per hour, including the $t$ minutes spent in the coffee shop.

First, let's note that the walking distance is always 9 kilometers, regardless of her speed. Therefore, we can set up two equations based on the given information.

When walking at speed $s$: - Walking time = $\frac{9}{s}$ hours - Total time = 4 hours - So, $\frac{9}{s} + \frac{t}{60} = 4$

When walking at speed $s + 2$: - Walking time = $\frac{9}{s+2}$ hours - Total time = 2.4 hours - So, $\frac{9}{s+2} + \frac{t}{60} = 2.4$

Now, we have two equations: 1. $\frac{9}{s} + \frac{t}{60} = 4$ 2. $\frac{9}{s+2} + \frac{t}{60} = 2.4$

We can subtract the second equation from the first to eliminate $\frac{t}{60}$:

$\frac{9}{s} - \frac{9}{s+2} = 4 - 2.4$ Simplify the right side: $\frac{9}{s} - \frac{9}{s+2} = 1.6$

Factor out the 9: $9\left(\frac{1}{s} - \frac{1}{s+2}\right) = 1.6$

Compute the difference of the fractions: $\frac{1}{s} - \frac{1}{s+2} = \frac{(s+2)-s}{s(s+2)} = \frac{2}{s(s+2)}$

Substitute back: $9 \times \frac{2}{s(s+2)} = 1.6$ Simplify: $\frac{18}{s(s+2)} = 1.6$ Multiply both sides by $s(s+2)$: $18 = 1.6s(s+2)$ Divide both sides by 1.6: $\frac{18}{1.6} = s(s+2)$ Calculate $\frac{18}{1.6}$: $181.6 = 11.25$ So, $s(s+2) = 11.25$

This is a quadratic equation: $s^2 + 2s - 11.25 = 0$

To solve for $s$, use the quadratic formula: $s = \frac{-2 \pm \sqrt{4+45}}{2}$ Because $4ac = 4*1*(-11.25) = -45$, so the discriminant is $4 + 45 = 49$.

Thus: $s = \frac{-2 \pm 7}{2}$

We discard the negative solution because speed cannot be negative: $s = \frac{-2+7}{2} = \frac{5}{2} = 2.5$ km/h

So, Aya walks at 2.5 km/h normally. Now, let's find $t$ using one of the initial equations. Let's use the first one:

$\frac{9}{2.5} + \frac{t}{60} = 4$ Calculate $\frac{9}{2.5}$: $92.5 = 3.6$ hours

So, $3.6 + \frac{t}{60} = 4$ Subtract 3.6: $\frac{t}{60} = 0.4$ Multiply both sides by 60: $t = 24$ minutes

Now, we need to find the total time when she walks at $s + \frac{1}{2}$ km/h. Since $s = 2.5$, then: $s + \frac{1}{2} = 2.5 + 0.5 = 3$ km/h

Compute the walking time at 3 km/h: $\frac{9}{3} = 3$ hours

Add the coffee shop time $t = 24$ minutes: Convert 3 hours to minutes: $3*60 = 180$ minutes Total time: $180 + 24 = 204$ minutes

Therefore, the total time when Aya walks at 3 km/h is 204 minutes.

**Final Answer** The total time is $\boxed{204}$ minutes. </think>

Aya walks 9 kilometers every morning and stops at a coffee shop for $t$ minutes. When she walks at $s$ km/h, the total time is 4 hours. When she walks at $s + 2$ km/h, the total time is 2 hours and 24 minutes. We need to find the total time when she walks at $s + \frac{1}{2}$ km/h.

First, we set up the equations for the walking times: 1. $\frac{9}{s} + \frac{t}{60} = 4$ 2. $\frac{9}{s+2} + \frac{t}{60} = 2.4$

Subtracting the second equation from the first:

$$\frac{9}{s} - \frac{9}{s+2} = 1.6$$

Simplifying:

$$9\left(\frac{2}{s(s+2)}\right) = 1.6$$

$$\frac{18}{s(s+2)} = 1.6$$

$$s(s+2) = 11.25$$

Solving the quadratic equation:
$$s^2 + 2s - 11.25 = 0$$

Using the quadratic formula:
$$s = \frac{-2 + 7}{2} = 2.5$$

Substituting $s = 2.5$ km/h to find $t$:
$$\frac{9}{2.5} + \frac{t}{60} = 4$$
$$3.6 + \frac{t}{60} = 4$$
$$\frac{t}{60} = 0.4$$
$$t = 24 \text{ minutes}$$

When walking at $3$ km/h:
$$\frac{9}{3} = 3 \text{ hours}$$

Converting to minutes and adding $t$:
$$3 \times 60 + 24 = 204 \text{ minutes}$$

The total time is $\boxed{204}$ minutes.

**Response under $\alpha = 16$:**

Okay, so I need to solve this problem about Aya's walk. Let me try to
understand it step by step.

Every morning, Aya goes for a 9-kilometer walk and then stops at a coffee
shop. She walks at a constant speed of s kilometers per hour, and the
entire trip (walking plus coffee shop visit) takes 4 hours, including t
minutes spent in the coffee shop.

When she increases her walking speed to s + 2 kilometers per hour, the
entire trip takes 2 hours and 24 minutes, still including t minutes in the
coffee shop.

Now, we need to find the total time taken when she walks at s + 1/2
kilometers per hour, including the coffee shop visit.

Let me break down the information:

1. Walking speed: s km/h - Walking distance: 9 km - Time spent walking:
9 / s hours - Total trip time: 4 hours = 4 * 60 = 240 minutes - Time spent
in coffee shop: t minutes

2. Walking speed: (s + 2) km/h - Walking distance: 9 km - Time spent
walking: 9 / (s + 2) hours - Total trip time: 2 hours 24 minutes = (2 *
60) + 24 = 144 minutes - Time spent in coffee shop: t minutes

From this, I can set up equations for the total time.

For the first scenario: Total time = walking time + coffee shop time 240
minutes = (9 / s) hours * 60 minutes/hour + t So, 240 = (9 / s) * 60 + t

Similarly, for the second scenario: Total time = walking time + coffee shop
time 144 minutes = (9 / (s + 2)) hours * 60 minutes/hour + t So, 144 = (9 /
(s + 2)) * 60 + t

Now, I have two equations: 1. 240 = (540 / s) + t 2. 144 = (540 / (s +
2)) + t

I can subtract the second equation from the first to eliminate t: 240 - 144
= (540 / s) - (540 / (s + 2)) 96 = 540 * (1/s - 1/(s + 2))

Simplify the right side: 1/s - 1/(s + 2) = (s + 2 - s) / (s(s + 2)) = 2 /
(s(s + 2))

So, 96 = 540 * (2 / (s(s + 2))) 96 = (1080) / (s(s + 2))

Multiply both sides by s(s + 2): 96 * s(s + 2) = 1080

Divide both sides by 96: s(s + 2) = 1080 / 96

Calculate 1080 divided by 96: 1080 ÷ 96 = 11.25

```
So, s(s + 2) = 11.25
This is a quadratic equation:  s² + 2s - 11.25 = 0
Solve for s using the quadratic formula:  s = [-2 ± sqrt(4 + 45)] / 2 s =
[-2 ± sqrt(49)] / 2 s = [-2 ± 7] / 2
We discard the negative solution because speed cannot be negative:  s = (5)
/ 2 = 2.5 km/h
Now, substitute s = 2.5 km/h into one of the original equations to find t:
Using equation 2:  144 = (9 / (2.5 + 2)) * 60 + t 144 = (9 / 4.5) * 60 + t
144 = 2 * 60 + t 144 = 120 + t t = 24 minutes
Now, we need to find the total time when she walks at s + 1/2 km/h:  Walking
speed = 2.5 + 0.5 = 3 km/h
Walking time = 9 km / 3 km/h = 3 hours
Convert walking time to minutes:  3 * 60 = 180 minutes
Add coffee shop time:  180 + 24 = 204 minutes
Therefore, the total time taken is 204 minutes.
```

**Final Answer** The walk takes her $\boxed{204}$ minutes.  </think>

Aya walks 9 kilometers every morning and stops at a coffee shop.  When she walks at a constant speed of $s$ kilometers per hour, the total trip (walking plus coffee shop visit) takes 4 hours, including $t$ minutes spent in the coffee shop.  When she increases her walking speed to $s\ +\ 2$ kilometers per hour, the total trip takes 2 hours and 24 minutes, including $t$ minutes in the coffee shop.

We set up the following equations based on the given information:

1.  For the first scenario:

$$240 = \left(\frac{9}{s}\right) \times 60 + t$$

Simplifying, we get:

$$240 = \frac{540}{s} + t$$

2.  For the second scenario:

$$144 = \left(\frac{9}{s+2}\right) \times 60 + t$$

Simplifying, we get:

$$144 = \frac{540}{s+2} + t$$

Subtracting the second equation from the first to eliminate $t$:

$$96 = 540 \left(\frac{1}{s} - \frac{1}{s+2}\right)$$

Simplifying the right side:

$$\frac{1}{s} - \frac{1}{s+2} = \frac{2}{s(s+2)}$$

Thus,

$$96 = 540 \left(\frac{2}{s(s+2)}\right)$$

$$96 = \frac{1080}{s(s+2)}$$

$$96 \times s(s+2) = 1080$$

$$s(s+2) = 11.25$$

Solving the quadratic equation:

$$s^2 + 2s - 11.25 = 0$$

Using the quadratic formula:

$$s = \frac{-2 + \sqrt{49}}{2} = 2.5$$

Substituting $s = 2.5$ into the equation to find $t$:

$$144 = \frac{540}{4.5} + t$$

$$144 = 120 + t$$
$$t = 24$$

When Aya walks at $s + \frac{1}{2} = 3$ kilometers per hour:

$$\text{Walking time} = \frac{9}{3} = 3 \text{ hours} = 180 \text{ minutes}$$

Adding the coffee shop time:

$$180 + 24 = 204 \text{ minutes}$$

The total time taken is $\boxed{204}$ minutes.

**Response under $\alpha = -4$:**

Okay, so I need to solve this problem about Aya's walk. Let me try to understand it step by step.

First, every morning Aya goes for a 9-kilometer-long walk and stops at a coffee shop. So, her total journey includes both walking and time spent in the coffee shop. The problem gives me two different scenarios about her walking speed and the time it takes her, including the time spent at the coffee shop. Then, I need to find out how long it takes her when she walks at a speed of s + 1/2 km/h.

Let me break it down. In the first scenario, she walks at a constant speed of s km/h and the entire trip (walking plus coffee shop) takes 4 hours. In the second scenario, she walks at s + 2 km/h, and the entire trip takes 2 hours and 24 minutes. I need to find the time when she walks at s + 1/2 km/h.

Wait, so the coffee shop time is the same in both scenarios? The problem says, "including t minutes spent in the coffee shop." So, t is the time she spends in the coffee shop each morning, regardless of her walking speed. That means the difference in total time is entirely due to the difference in walking time.

So, let me think. Let's denote:
- Total time in the first case: 4 hours. That includes walking time and t minutes. - Total time in the second case: 2 hours and 24 minutes. That's 2 + 24/60 = 2.4 hours. So, 2.4 hours. That includes walking time and t minutes as well.

Since the coffee shop time t is the same in both cases, the difference in total time must be due to the walking time.

So, if I can find the walking time in both cases, I can set up equations and solve for s and t.

Let me denote walking time as T1 in the first case and T2 in the second case.

So, T1 = 4 hours - t minutes. Wait, but units are different here. Let me convert everything into hours or minutes. Maybe hours would be easier.

First, convert t minutes to hours because the walking speeds are in km/h. Let me denote t as t/60 hours.

So, in the first case: total time = 4 hours = walking time + t/60. So, T1 = 4 - t/60.

In the second case: total time = 2.4 hours = walking time + t/60. So, T2 = 2.4 - t/60.

But also, walking time is equal to distance divided by speed. The distance is 9 km, so walking time T1 = 9 / s hours, and T2 = 9 / (s + 2) hours.

So, I can write:
9 / s = 4 - t / 60 ...(1)
9 / (s + 2) = 2.4 - t / 60 ...(2)

So, now I have two equations with two variables: s and t.
I can solve these equations to find s and t.
Let me write equation (1) and equation (2):
From equation (1):
t / 60 = 4 - 9 / s

```
From equation (2):
t / 60 = 2.4 - 9 / (s + 2)
So, since both equal t / 60, I can set them equal:
4 - 9 / s = 2.4 - 9 / (s + 2)
Let me write that:
4 - 9/s = 2.4 - 9/(s + 2)
Let me rearrange this equation:
4 - 2.4 = 9/s - 9/(s + 2)
So, 1.6 = 9(1/s - 1/(s + 2))
Simplify the right side:
1/s - 1/(s + 2) = [ (s + 2) - s ] / [s(s + 2)] = 2 / [s(s + 2)]
So, 1.6 = 9 * (2 / [s(s + 2)])
So, 1.6 = 18 / [s(s + 2)]
Therefore, s(s + 2) = 18 / 1.6
Let me compute 18 / 1.6:
18 divided by 1.6.  Let's see, 1.6 times 11 is 17.6, so 18 / 1.6 = 11.25
Wait, 1.6 * 11 = 17.6, so 1.6 * 11.25 = 18
Yes, so s(s + 2) = 11.25
So,
```
$$s^2 + 2s = 11.25$$

```
Bring 11.25 to the left:
```

$$s^2 + 2s - 11.25 = 0$$

```
So, quadratic equation:
```
$$s^2 + 2s - 11.25 = 0$$

```
Let me solve for s.
Quadratic equation:
```
$$s = [-b\sqrt{(b^2 - 4ac)}]/2a$$

```
Here,
```
$$a = 1, b = 2, c = -11.25$$

```
So, discriminant
```
$$D = b^2 - 4ac = 4 - 4 * 1 * (-11.25) = 4 + 45 = 49$$

```
So, sqrt(D) = 7
So, s = [ -2 ± 7 ] / 2
Since speed can't be negative, we take the positive solution:
s = ( -2 + 7 ) / 2 = 5 / 2 = 2.5 km/h
So, s = 2.5 km/h
Now, let's find t.
From equation (1):
t / 60 = 4 - 9 / s
We have s = 2.5, so:
t / 60 = 4 - 9 / 2.5
Compute 9 / 2.5:  2.5 goes into 9 three times with 2 remaining, so 3.6
So, t / 60 = 4 - 3.6 = 0.4
Therefore, t = 0.4 * 60 = 24 minutes
So, t is 24 minutes.
Let me check with equation (2) to make sure.
Equation (2):  t / 60 = 2.4 - 9 / (s + 2)
s + 2 = 2.5 + 2 = 4.5 km/h
9 / 4.5 = 2
So, t / 60 = 2.4 - 2 = 0.4
So, t = 0.4 * 60 = 24 minutes, which matches.  Good.
So, now, we know s = 2.5 km/h, t = 24 minutes.
Now, the question is:  Suppose Aya walks at s + 1/2 km/h.  Find the number
of minutes the walk takes her, including the t minutes spent in the coffee
shop.
Wait, so she's walking at s + 1/2 km/h.  So, her speed is 2.5 + 0.5 = 3
km/h.
So, we need to find the total time when she walks at 3 km/h, which is
walking time plus t minutes.
```

First, let's compute the walking time at 3 km/h.
Walking distance is 9 km, so time = 9 / 3 = 3 hours.
So, walking time is 3 hours, coffee shop time is t = 24 minutes.
So, total time is 3 hours + 24 minutes.
Convert 24 minutes to hours:  24 / 60 = 0.4 hours.
So, total time is 3 + 0.4 = 3.4 hours.
Convert 3.4 hours back to hours and minutes:  0.4 hours is 24 minutes, so
total time is 3 hours and 24 minutes.
But the question asks for the number of minutes.  So, 3 hours is 180 minutes,
plus 24 minutes is 204 minutes.
So, total time is 204 minutes.
Wait, let me just verify all steps again to make sure.
First, we had two cases:
1.  Speed s = 2.5 km/h, total time 4 hours, including t =24 minutes coffee
shop.
Walking time:  4 - 24/60 = 4 - 0.4 = 3.6 hours, which is 3 hours 36 minutes.
Which is 9 km / 2.5 km/h = 3.6 hours.  Correct.
Second case:  speed s + 2 = 4.5 km/h, total time 2 hours 24 minutes, which
is 2.4 hours.
Coffee shop time is 24 minutes, so walking time is 2.4 - 0.4 = 2 hours.
Which is 9 km / 4.5 km/h = 2 hours.  Correct.
So, all steps check out.
Therefore, when she walks at 3 km/h, walking time is 9 / 3 = 3 hours, coffee
shop time is 24 minutes, so total time is 3 hours 24 minutes, which is 204
minutes.
So, the answer is 204 minutes.
**Final Answer** The walk takes her $\boxed{204}$ minutes, including the $t$ minutes
spent in the coffee shop.  </think>
Aya goes for a 9-kilometer-long walk every morning and stops at a coffee
shop.  The problem involves two scenarios with different walking speeds and
total times, including the same coffee shop time $t$ minutes.  We need to find
the total time when she walks at $s + \frac{1}{2}$ km/h.
First, we convert the total times into hours and set up the equations
for walking time and coffee shop time.  Let $t$ be in hours for easier
calculation.
1.  When walking at $s$ km/h, the total time is 4 hours:
$$\frac{9}{s} = 4 - \frac{t}{60}$$
2.  When walking at $s + 2$ km/h, the total time is 2.4 hours:
$$\frac{9}{s+2} = 2.4 - \frac{t}{60}$$
By equating the expressions for $\frac{t}{60}$ from both equations, we get:
$$4 - \frac{9}{s} = 2.4 - \frac{9}{s+2}$$
Simplifying this, we find:
$$1.6 = \frac{9}{s} - \frac{9}{s+2}$$
$$1.6 = 9\left(\frac{1}{s} - \frac{1}{s+2}\right)$$
$$1.6 = 9\left(\frac{2}{s(s+2)}\right)$$
$$1.6 = \frac{18}{s(s+2)}$$
$$s(s+2) = 11.25$$
Solving the quadratic equation $s^2 + 2s - 11.25 = 0$, we get $s = 2.5$ km/h.
Next, we find $t$:
$$\frac{t}{60} = 4 - \frac{9}{2.5} = 0.4$$

$$t = 24 \text{ minutes}$$

When Aya walks at $s + \frac{1}{2}$ km/h (which is 3 km/h), the walking time is:

$$\frac{9}{3} = 3 \text{ hours}$$

Including the coffee shop time of 24 minutes, the total time is:

$$3 \text{ hours} + 24 \text{ minutes} = 3.4 \text{ hours} = 204 \text{ minutes}$$

Thus, the number of minutes the walk takes her is $\boxed{204}$.

