# OpenReview forum: "Controlling Thinking Speed in Reasoning Models"
_NeurIPS.cc/2025/Conference — NeurIPS 2025 spotlight_

### Official Review · Reviewer_duWd · 2025-07-01

**Clarity:** 3
**Significance:** 3
**Originality:** 3
**Rating:** 5
**Confidence:** 3

**Summary:**

This paper introduces a dynamic thinking speed adjustment method for Large Reasoning Models. By analyzing two types of model outputs, i.e., fast, intuition-based answers and slower, reasoning-based responses, the authors propose summarizing a "reading vector" from fast and slow response samples to intervene the model's inference behavior. They first demonstrate that slowing down inference leads to improved reasoning performance. Then, they introduce a heuristic-based dynamic speed control algorithm that enables the model to switch between intuitive and fast reasoning states, achieving comparable results with improved computational efficiency.

**Questions:**

1. Would it be possible to visualize the $\alpha$ values as a time series over the course of text generation? This could help assess whether the proposed sliding-window algorithm is performing effectively and provide insights into how the control signal influences generation in real time.

2. What happens if $\alpha$ becomes too large or too small? How can we ensure that such variations do not disrupt the semantic content of the generation? In other words, is the control direction disentangled from the actual content or the solution being generated?

3. Some prior work introduces special tokens like "think" or "pause" to encourage slower, more deliberate inference. How does your approach compare to these methods, and are there potential complementarities? (E.g., "Think before you speak: Training Language Models With Pause Tokens" @ ICLR 2024)

**Ethical Concerns:**

["NO or VERY MINOR ethics concerns only"]

**Final Justification:**

I thank the authors for providing the additional experiments. Some more suggestions are raised, and I will maintain my score.

**Limitations:**

The authors have adequately discussed the limitations in the appendices.

**Quality:**

3

**Strengths And Weaknesses:**

### Strengths:
The paper is easy to follow, and it is valuable that it tackles the challenge of controllability across different thinking modes by directly interpreting and intervening in the model's internal architecture. The problem is well-defined within a clear scope, and the experiments demonstrate the effectiveness of the proposed method. The work also raises interesting questions about whether classical tools such as PCA and information theory could further aid in understanding the "flow" of large language models.

### Weaknesses:
The proposed method, in its spirit,, appears to be closely related to citation [41]. It may be helpful to better motivate this connection by introducing more of the high-level ideas behind [41], rather than focusing only on methodological overlap. In other words, does the theory in [41] help explain or justify the proposed heuristic intervention approach? If so, drawing that connection more clearly could strengthen the argument of this paper. (Please correct me if I’m mistaken.)

As the authors also acknowledge, the current method is largely heuristic. Some theoretical findings are expected in future research.

---

> ### Author Rebuttal · Authors · 2025-07-31
>
> We are pleased that you find our work valuable, interesting, and our methods effective. We hope the responses below address your concerns.
>
> ## W1
> Q: Does the theory in [41] help explain or justify the proposed heuristic intervention approach?
>
> **A**: We thank the reviewer for the suggestion to clarify the connection between our work and citation [41] (Representation Engineering, RepE). We agree that RepE inspires and provides a theoretical foundation for our method. But we also see our work as a valuable extension of RepE-like framework.
>
> First, our core motivation for manipulating LLMs' thinking speed stems from the hypothesis that **reasoning styles** (fast/slow thinking) belong to **high-level, abstract cognitive functions** within LLMs. This hypothesis aligns with the top-down, representation-centered view in RepE to identify such functions, in contrast to bottom-up circuit-level analysis. Moreover, our observations of keyword-triggered behaviors further support that different thinking modes are organized along **directional subspaces** in the model's representation space. This insight justifies our use of PCA to identify steering directions.
>
> While our method builds upon RepE's core principles, we significantly extend its framework both **theoretically and functionally**. Our early experiments revealed that static, global interventions could only govern model responses toward either fast (efficient) or slow (accurate) thinking modes. This limitation constrains our goal of enhancing both accuracy and efficiency simultaneously.
>
> We attribute these constraints to a _fundamental limitation_ in existing RepE-based approaches: their interventions operate along a _single functional_ axis, applying uniform behavior controls across entire generations. To address this, one of our key contributions is the introduction of **adaptive control**, which reframes the traditional intervention paradigm into a **two-dimensional** control task by incorporating the **temporal dimension**. Specifically, we make the intervention reasoning-aware and context-specific by dynamically adjusting the strength of representation control based on the model's evolving internal reasoning state during generation.
>
> This temporal adaptivity fundamentally distinguishes our work by focusing on **when and how models should think** (dynamic strategy control) rather than _what they should think (static behavior control)_, thereby bridging RepE into the domain of cognitive process control.
>
>
> ## W2
> Q: The current method is largely heuristic. Some theoretical findings are expected in future research.
>
> **A**: Though our current methods, such as sliding-window-based adaptive control, remain heuristic, we believe that the core motivation for introducing temporal adaptivity, along with the success of our experimental results and analyses, **provides valuable insights** into the interpretability of LLMs' reasoning flow and may inspire future work to revisit the representation-control framework.
>
> ## Q1
> Q: Would it be possible to visualize the $\alpha$ values as a time series over the course of text generation?
>
> **A**: We provide the following statistical analysis to elucidate our adaptive control method's dynamics. Since the main function of our adaptive control methods is to force LRMs to switch to slow-thinking mode (i.e., set $\alpha$ to negative values) when encountering difficult reasoning, we specifically study **how the frequencies of these switches evolve over time**. We first divide the model's responses into 3 temporal segments: **start** (first 25%), **middle** (25%-75%), and **end** (last 25%). We measure switching frequency by calculating the average token interval between these switches (where shorter intervals indicate more frequent switching). Below we present results from DeepSeek-R1-Distill-Qwen-7B.
> | **Dataset** | | | Avg. token interval between switches|
> |-|-|-|-|
> ||Start|Mid|End|
> |AIME24|17.8|20.9|54.9|
> |MATH500|14.5|22.5|41.7|
> We found that:
> 1. The model tends to switch to slow-thinking mode most frequently at the beginning. This is reasonable, as LRMs typically exhibit complex and diverse reasoning behaviors at this stage, such as problem analysis, knowledge point recall, and solution plan development.
> 2. The middle segment shows the second-highest switching frequency, as the model begins problem-solving. This phase often involves intensive reasoning and reflection on trial-and-error attempts.
> 3. The switching frequency drastically decreases in the final segment. This is expected, as the model usually converges on its final solution and ceases to generate new thoughts.
> These patterns demonstrate that our control algorithm produces reasoning dynamics that closely align with human cognitive processes.
>
> To understand **how this control signal influences the generation**, we identify the top-5 most frequent tokens immediately after slow-thinking switches:
> ```
> [" no", " maybe", " let", " perhaps", " but"]
> ```
> These hesitation markers consistently signal **reflections** and **reconsiderations** in models' CoTs. For quantitative analysis, we include detailed case studies in Figure B1~3, which further validate our method's effectiveness in triggering crucial reasoning behaviors for accurate problem-solving. These findings will be included in our final version of paper.
> ## Q2
> Q: What happens if $\alpha$ becomes too large or too small? How can we ensure that such variations do not disrupt the semantic content of the generation?
>
> **A**: To study the effect of extreme values of steering intensity $\alpha$, we scale the absolute value of $\alpha$ when applying our thinking speed control to DeepSeek-R1-Distill-Qwen-7B on AIME24. The results are shown below:
> |  | $\alpha=64$  | 32 | 16 | 8 | 0 | -2 | -4 | -6 | -16 | -32 |
> |-|-|-|-|-|-|-|-|-|-|-|
> |**Acc**|0.0|6.2|41.7|53.3|52.5|53.7|55.4|52.9|39.6|0.8|
> |**Length**|Repetitive generation|1941.2|6232.5|8735.6|12451.2|14364.6|15144.8|15843.4|20241.7|Repetitive generation|
>
> The results show that when $|\alpha|$ becomes excessively large (e.g., $\alpha=64$ or $-32$), the generation quality degrades significantly, leading to semantic collapse and repetitive outputs.
>
> Regarding the second question, we find that fully disentangling reasoning style from semantic content is challenging. Our experiments in Q1 show that steering with negative $\alpha$ can induce reflective behavior, as evidenced by increased usage of tokens such as "no" and "but". However, we believe the underlying mechanism is more complex than simple token-level shifts, with **minimal semantic impact** in most cases. This is evidenced by (1) semantic disruption occurs only with extremely large values of $\alpha$, and (2) the accuracy-efficiency trade-off remains stable across a wide range of $\alpha$ values within reasonable bounds.
>
> The above observations suggest that for reasoning benchmarks, the critical requirement for effective intervention methods is preserving the **semantic correctness** of generated content. To further assess our intervention methods, we compare our thinking-speed vector to 2 baseline vectors:
> * **Hesitation vector**: To test whether our control vector merely increases the likelihood of hesitation tokens, we construct a "hesitation" vector using a modified stimulus pair. The negative stimulus remains the same (slow CoT), but the positive stimulus is generated by appending the word "Wait" to its paired negative stimulus. The rest of the vector construction process follows our original protocol. Notably, we observe that this vector leads to semantic collapse and repetition at a much lower intensity ( $\alpha' = 8$ ) .
> * **Correctness vector**: We also explore whether our control direction inadvertently overlaps with the model’s sense of solution correctness. We construct a "correctness" vector using paired correct and incorrect solutions from PRM800K as stimuli.
>
> We measure the cosine similarity between our control vector and these two baselines across the last 10 layers of LRMs. The results below suggest that our vector exhibits only minor overlap with the hesitation vector ($\approx 10^\circ$) and remains largely orthogonal to the correctness vector. This further suggests that our control direction is **not trivially token-driven and does not interfere with task correctness**.
> |  |  |  |  |  |  | Layer |  |  |  |  |
> |-|-|-|-|-|-|-|-|-|-|-|
> | **Vectors** |19|20|21|22|23|24|25|26|27|28|
> |Hesitation|81.5|82.5|82.7|79.3|79.4|78.2|79.7|80.9|80.8|85.0|
> | Correctness |94.3|94.4|93.0|90.8|90.5|93.3|92.7|95.5|93.3|96.7|
>
> ## Q3
> Q: Some prior work introduces special tokens like "think" or "pause". How does your approach compare to these methods, and are there potential complementarities?
>
> **A**: We thank the reviewer for pointing out the connection. The mechanisms behind their works and ours are quite different.
>
> The pause-token methods introduce explicit delays via special tokens, which requires _retraining the model_. They extend the thoughts of models by _explicitly increasing the number of internal hidden states_.
>
> In contrast, our vector-based intervention method introduces **no architectural or training modifications**. We operate on the LLM's representation space. Our approach is thus **lightweight, inference-time only, and generalizable** across diverse models without retraining.
>
> Despite these differences, we foresee a combination of our methods with these approaches. Since pause tokens offer additional computational bandwidth, our control signal could also be used to govern when and how that bandwidth is used. For models undergoing pause-token training, we can study the mechanisms for the trigger of pause tokens. This would provide more insights for studying the mechanism of reasoning flows.

---

> > ### Comment · Reviewer_duWd · 2025-08-05
> >
> > Thank you for the detailed rebuttal. I found the new material very interesting, and I believe it would strengthen the paper if these additions were incorporated into the final version. Based on our exchange, I would like to offer two additional suggestions:
> >
> > 1. Regarding W1: I am not questioning the novelty of the proposed approach. On the contrary, I believe a more thorough discussion of citation [41] would help clarify the conceptual foundation of your method. Without this, the proposed vector tuning strategy may come across as somewhat ad-hoc to readers unfamiliar with the background.
> >
> > 2. Regarding Q2: I understand there is a potential trade-off between semantic correctness and speed controllability, or perhaps the trade-off is not particularly significant. In either case, it would be helpful if the authors could provide a compact, quantitative summary of this relationship to better inform the reader.
> >
> > Thank you for providing the additional experiments, and I will maintain my score.

---

> > > ### Author Response · Authors · 2025-08-06
> > >
> > > We sincerely thank the reviewer for the kind follow-up and thoughtful suggestions. We're very encouraged that you found the new material interesting and appreciate your additional advice.
> > >
> > > * Regarding W1, we agree that it is important to provide background on the representational theory introduced in [41]. RepE proposes that abstract cognitive functions can be encoded as linear directions in activation space. We hypothesize that fast and slow thinking modes fall within this category, and our findings support this view: different reasoning styles are organized along distinct directional subspaces, and steering vectors corresponding to these directions can be extracted to modulate reasoning behavior. We will add this clarification to the final version of the paper.
> > >
> > > * Regarding Q2, we will include the experimental results from our rebuttal along with a concise summary, highlighting that:
> > >     * Steering remains effective and semantically stable across a wide range of $\alpha$ values, with semantic collapse only observed at extreme values (e.g., $\alpha=64$ or $\alpha=-32$).
> > >     * The steering vector for reasoning speed control remains largely orthogonal to directions associated with correctness judgment and hesitation token likelihood, suggesting a clean separation from semantic content and correctness.
> > >
> > > Thank you again for your constructive feedback. We will incorporate these suggestions to further improve the clarity and impact of the final version.

---

### Official Review · Reviewer_eb6B · 2025-07-02

**Clarity:** 3
**Significance:** 3
**Originality:** 3
**Rating:** 4
**Confidence:** 3

**Summary:**

This work proposes a method to control the thinking speed of large reasoning models while maintaining or even improving accuracy. It identifies differences in leading words between slow and fast thought processes and introduces a thinking speed control method based on learned representation differences between fast and slow response pairs. The method's effectiveness is demonstrated across various datasets and reasoning models. Additionally, an adaptive speed control method is proposed, leveraging problem difficulty estimation, which shows performance improvements over the non-adaptive approach.

**Questions:**

Could the authors provide additional results on the generality and transferability of both the leading words and the control vector? Furthermore, how do the results in Figure 5 compare to prompt engineering approaches, such as providing different leading words as illustrated in Figure 2? These points are well addressed by the authors in the responses.

**Ethical Concerns:**

["NO or VERY MINOR ethics concerns only"]

**Final Justification:**

I find this to be a technically solid paper. While I am uncertain about its practical applicability, I am inclined towards acceptance.

**Limitations:**

yes

**Paper Formatting Concerns:**

N.A,

**Quality:**

3

**Strengths And Weaknesses:**

The identification of distinctions in leading words between slow and fast thinking processes is interesting. However, it is unclear whether this distinction applies to other reasoning models (e.g., o1) or non-reasoning models, as it might be a side effect of a specific reinforcement learning protocol. The thinking speed control method is intriguing, particularly the existence of a vector in the representation space that adjusts thinking speed. Yet, the generalizability of this vector across other reasoning and non-reasoning models remains uncertain. The work would benefit from deeper investigation into these core phenomena to better understand the proposed approach's broader applicability.

---

> ### Author Rebuttal · Authors · 2025-07-31
>
> We are glad that you find our work interesting and our results intriguing. We hope our additional experiments and the responses below could address your concern.
> ## W1 & Q1-part1
> Q: Could the authors provide additional results on the generality and transferability of both the leading words and the control vector?
>
> **A**: We appreciate this valuable suggestion. To evaluate the generality of keyword-triggered behaviors and the transferability of our control vector, we conducted experiments across a diverse set of models, including both non-reasoning and reasoning models:
> * For non-reasoning models, we include `Qwen2.5-7B-Instruct` and `Llama3.1-8B-Instruct`, which represents **different architectures and training protocols**.
>  * For reasoning models: Since o1 is closed-source with limited API functionality, we instead evaluate the following open-source alternatives: `DeepSeek-R1-Distill-Llama-8B` and `nvidia/Llama-3.1-Nemotron-Nano-8B-v1`. Both models are Llama3.1-8B derivatives but employ **contrasting post-training approaches** (distillation vs. reinforcement learning), enabling direct comparison of RL effects. We additionally evaluate `GLM-Z1-9B-0414`, which features a **distinct architecture** from both Qwen and Llama families and is trained with alternative RL protocols, to further assess cross-architectural transferability.
>
> We first evaluate the effectiveness of leading words for triggering slow and fast thinking modes across these models using the MATH500 benchmark.
>
> For non-reasoning models, we treat their original outputs as fast-thinking responses and attempt to induce longer reasoning chains using the most common slow-thinking trigger identified in LRMs (Figure 2), i.e., the word "Okay". The results shown below demonstrate that this trigger fails to consistently extend the models' reasoning processes. This outcome is expected, as these non-reasoning models lack specialized long CoT training and have limited exposure to long reasoning data, making their slow thinking capabilities difficult to activate through simple prompting.
> | **Model** | **Slow-thinking**(Pass@1 (%) / Output Length) | **Fast-thinking** (Pass@1 (%) / Output Length)|
> | --- | --- | --- |
> | Qwen2.5-7B-Instruct | 74.8 / 654.0 | 75.9 / 655.2 |
> | Llama-3.1-8B-Instruct | 43.9 / 1188.7 | 41.5 / 702.2 |
>
> For reasoning models, we treat their original MATH500 responses as slow-thinking outputs and elicit fast-thinking responses using the trigger word "To" (consistent with our main paper). The results below demonstrate **consistent keyword-triggered behavior** across diverse LRMs, showing robustness to variations in model architecture and training protocols.
> | **Model** | **Slow-thinking**(Pass@1 (%) / Output Length)|**Fast-thinking** (Pass@1 (%) / Output Length) | **Relative Differences**|
> | --- | --- | --- | --- |
> | Llama-3.1-Nemotron-Nano-8B-v1 | 93.7 / 3893.4 | 91.6 / 2760.6 | -2.2% / -29.1% |
> | DeepSeek-R1-Distill-Llama-8B | 88.2 / 3694.1 | 63.8 / 878.7 | -27.7% / -76.2% |
> | GLM-Z1-9B-0414 | 96.0 / 2847.1 | 91.7 / 1691.8 | -4.5% / -40.6% |
>
> Next, we evaluate the transferability of our control vector extraction method and the generalizability of the resulting thinking speed control across all models. After extracting the control vector from these models using the same method in Section 3, we use the extracted vector to manipulate the models' reasoning behaviors using different steering intensities on AIME24 and MATH500. The results are shown as follows. Despite the differences on the keyword triggering behaviors between non-reasoning and reasoning models, the scaling effects introduced by our intervention methods **generalized across all types of models**, demonstrating the **transferability** of our methods. Notebaly, to extract the models' representations for different thinking modes, we reuse the slow and fast CoTs generated by DeepSeek-R1-Distill-Qwen-7B across all tested models. The effectiveness based on this stimulus sharing further highlights that the ability to think in both modes should be an inherent ability shared across all models, which is more likely unrelated to any specific training protocols or certain model families.
>
> | Models | AIME 24 ( Pass@1 (%) / Output Length )|  |  |  |  |
> | --- | --- | --- | --- | --- | --- |
> |  | $\alpha=8$ | $\alpha=4$ | $\alpha=0$ | $\alpha=-2$ | $\alpha=-4$ |
> | **Non-reasoning** |  |  |  |  |  |
> | Llama3.1-8B-Instruct | **8.3** / 783.8 | 5.4 / 1293.2 | 6.8 / 3501.2 | 6.7 / 3787.6 | **7.1** / 4748.7 |
> | Qwen2.5-7B-Instruct | 9.2 / 1574.9 | **10.4** / 1841.0 | 9.6  /  1886.7 | **11.7** / 1959.4 | **10.8** / 3177.7 |
> | **Reasoning** |  |  |  |  |  |
> | Llama-3.1-Nemotron-Nano-8B-v1 | 47.1 / 8072.1 | 62.1/ 10719.1 | 62.9 / 11044.5 | **65.0** / 11682.1 | **63.3** / 13414.8 |
> | DeepSeek-R1-Distill-Llama-8B | 35.0 / 10589.4 | 42.1 / 11164.8 | 50.0 / 13398.3 | **53.3** / 14709.1 | **51.7** / 15776.1 |
> | GLM-Z1-9B-0414 | 65.0 / 7870.0 | 67.1 / 8346.8 | 67.5 / 8883.45 | **71.3** / 9585.57 | **72.1** / 10150.1 |
>
> | Models | MATH500 ( Pass@1 (%) / Output Length ) |  |  |  |  |
> | --- | --- | --- | --- | --- | --- |
> |  | $\alpha=8$ | $\alpha=4$ | $\alpha=0$ | $\alpha=-2$ | $\alpha=-4$ |
> | **Non-reasoning** |  |  |  |  |  |
> | Qwen2.5-7B-Instruct | 72.9 / 534.6 | 74.0 / 544.2 | 75.9 / 655.2 | **76.2** / 700.3 | 75.4 / 868.0 |
> | Llama3.1-8B-Instruct | **41.6** / 541.8 | **42.9** / 599.4 | 41.5 / 702.2 | **42.8** / 773.9 | 39.5 / 815.2 |
> | **Reasoning**  |  |  |  |  |  |
> | Llama-3.1-Nemotron-Nano-8B-v1 | 79.4 / 2117.6 | 87.7 / 3091.64 | 93.7 / 3893.4 | **94.3** / 4118.9 | **94.3** / 4303.09 |
> | DeepSeek-R1-Distill-Llama-8B | 75.9 / 2650.7 | 85.1 / 2895.3 | 88.2 / 3694.1 | **89.6** / 4241.2 | **90.2** / 4829.1 |
> | GLM-Z1-9B-0414 | 94.3 / 2215.0 | **96.2** / 2439.9 | 96.0 / 2847.1 | **96.5** / 3023.3 | **96.4** / 3265.5 |
>
> To summarize, although keyword-triggering behaviors may vary across models, the ability to switch between slow and fast thinking is broadly present **across a wide range of LLMs** and can be effectively manipulated using our intervention methods. We therefore believe our approach has broad applicability. Moreover, we hope that the insights underlying our methods and experimental findings will offer valuable contributions to future research and deepen our understanding of LLMs' internal reasoning processes.
>
> ## Q1-part2
> Q: How do the results in Figure 5 compare to prompt engineering approaches, such as providing different leading words as illustrated in Figure 2?
>
> **A**: Thank you for this valuable suggestion. We compared the Figure 5 results against three prompt engineering baselines:
> * Prompt1[1]:
> ```
> <|User|>[instruction]\nAnswer after a short amount of thinking. Do not spend excessive time double-checking your work.<|Assistant|><think>\n
> ```
> * Prompt2 (Fast-thinking):
> ```
> <|User|>[instruction]<|Assistant|><think>\nTo
> ```
> * Prompt3 [2]:
> ```
> <|User|>[instruction]<|Assistant|><think>\nOkay I have finished thinking.\n</think>\n
> ```
> Due to figure restrictions during this rebuttal period, we present the performance under each prompt strategy alongside the two closest intervention result points from Figure 5. We experiment with DeepSeek-R1-Distill-Qwen-32B and report the results as follow:
>
> *   MATH500
>
>
> | **Method** | **Length (x-axis)** | **Pass@1 (y-axis)** | **Token Efficiency (Pass@1 / Length, Slope,** %**)** |
> | --- | --- | --- | --- |
> | $\alpha=0$ | 2947.3 | 94.2 | 3.20 |
> | $\alpha=4$ | 2088.7 | 92.0 | 4.40 |
> | Prompt 1 | 2079.7 | 91.3 | 4.39 |
> | $\alpha=8$ | 1495.8 | 87.6 | 5.86 |
> | $\alpha=12$ | 1034.5 | 81.4 | 7.87 |
> | Prompt 2 | 967.1 | 79.1 | 8.18 |
> | $\alpha=16$ | 720.5 | 74.3 | 10.31 |
> | Prompt 3 | 641.7 | 79.8 | 12.43 |
>
>
> *   AIME24
>
> | **Method** | **Length (x-axis)** | **Pass@1 (y-axis)** | **Token Efficiency (Pass@1 / Length, Slope,** ‰**)** |
> | --- | --- | --- | --- |
> | $\alpha=0$ | 10679.6 | 69.2 | 6.47 |
> | Prompt 1 | 9510.7 | 65.4 | 6.87 |
> | $\alpha=4$ | 8000.1 | 60.8 | 7.60 |
> | $\alpha=12$ | 3893.7 | 38.3 | 9.84 |
> | Prompt 3 | 3436.4 | 26.2 | 7.62 |
> | Prompt 2 | 3076.3 | 27.1 | 8.81 |
> | $\alpha=16$ | 1784.1 | 21.7 | 12.16 |
>
> When plotted on an x-y plane, the results from these prompt engineering methods **generally fall below the scaling curve** generated by our speed control approach. Moreover, our intervention method offers smoother and more flexible accuracy-efficiency trade-offs, demonstrating clear advantages over traditional prompt engineering techniques.
>
> ---
> 1. s1: Simple test-time scaling
> 2. Reasoning Models Can Be Effective Without Thinking

---

> > ### Comment · Reviewer_eb6B · 2025-08-04
> >
> > Regarding the experiments of transferability, I find a reduced effect in other models. Is it possible to discuss about the relation between the optimal steering vectors of different models? Also, is there side effect by tuning \alpha to a VERY large scale in order to compensate for a reduced effect?

---

> > > ### Author Response · Authors · 2025-08-05
> > >
> > > Thank you for your valuable feedback. We hope our following explanations and experiments could address your concerns.
> > >
> > > ## Q1
> > > > Q: Is it possible to discuss about the relation between the optimal steering vectors of different models?
> > >
> > > **A**: Yes, we hypothesize that one way to improve the quality of the steering vectors for different models is to **use self-generated stimuli** during representation extraction. This approach allows each LLM’s unique internal representations of different thinking modes to be better captured.
> > >
> > >
> > > In our initial experiments, we demonstrate cross-model transferability of our steering methods by using the fast and slow CoTs generated exclusively by `DeepSeek-R1-Distill-Qwen-7B` (DS-QW-7B) for representation extraction. This approach yields consistent scaling effects under steering across all tested models.
> > > To further enhance steering performance, we examine a model-specific variant by replacing the stimuli for each model with self-generated fast and slow thinking data.
> > > The results on `MATH500` are shown as follows:
> > >
> > >
> > > | Models | Stimuli Source |  |  | MATH500 ( Pass@1 (%) / Output Length ) |  |  |
> > > | --- | --- | --- | --- | --- | --- | --- |
> > > |   |   | $\alpha=8$ | $\alpha=4$ | $\alpha=0$ | $\alpha=-2$ | $\alpha=-4$ |
> > > | Llama-3.1-Nemotron-Nano-8B-v1 | DS-QW-7B | 79.4 / 2117.6 | 87.7 / 3091.64 | 93.7 / 3893.4 | 94.3 / 4118.9 | 94.3 / 4303.09 |
> > > | Llama-3.1-Nemotron-Nano-8B-v1 | **Self-generated** | **80.4** / **1983.6** | 87.6 / **2813.8** | 93.7 / 3893.4 | **94.3** / **4237.0** | **94.6** / **4802.8** |
> > > | GLM-Z1-9B-0414 | DS-QW-7B | 94.3 / 2215.0 | 96.2 / 2439.9 | 96.0 / 2847.1 | 96.5 / 3023.3 | 96.4 / 3265.5 |
> > > | GLM-Z1-9B-0414 | **Self-generated** | 93.9 / **1981.8** | 94.7 / **2303.5** | 96.0 / 2847.1 | 96.4 / **3305.4** | **96.5** / **3851.9** |
> > >
> > > As shown above, using models' self-generated fast and slow CoTs as stimuli leads to a **stronger steering effect** on its thinking budgets, evidenced by greater variance in response lengths across different $\alpha$ values. We also observe improved steering quality, reflected in the generally higher accuracies under thought-expansion settings ($\alpha < 0$).
> > >
> > > We also acknowledge that additional factors, such as the quality of the original dataset (e.g., its difficulty) used for slow and fast stimulus sampling, as well as the criteria for selecting representative fast and slow samples, may influence the steering vector's quality. Exploring these variables is a promising direction for future work.
> > >
> > > However, we view our work as a first step toward **revealing the existence of fast and slow thinking modes** in LLMs and **unlocking new opportunities for reasoning strategies** that leverage these modes to control models' reasoning behaviors.
> > > Exploring the underlying mechanisms and formal relationships between optimal steering directions across models remains a promising direction for future research.
> > >
> > > ## Q2
> > > > Q: Is there side effect by tuning $\alpha$ to a VERY large scale in order to compensate for a reduced effect?
> > >
> > > **A**: Yes, but such side effects only emerge when $|\alpha|$ is pushed to extreme values (e.g., $\alpha = 64$ or $\alpha=-32$). As shown in our responses to Reviewers kUJX and duWd, we systematically scale $\alpha$ and observe that the steering effect remains **stable and effective across a wide range of values**. Only at the extreme values of $|\alpha|$ does the model begin to lose semantic coherence, leading to repetitive or degenerate outputs.
> > > This behavior is expected, as $\alpha$ serves as a hyperparameter and, like many others in machine learning, exhibits an effective operating range beyond which model behavior can become unstable.
> > >
> > > |  | $\alpha=64$ | 32 | $\alpha=16$ | $\alpha=8$ | $\alpha=0$ | $\alpha=-2$ | $\alpha=-4$ | $\alpha=-6$ | $\alpha=-16$ | $\alpha=-32$ |
> > > | --- | --- | --- | --- | --- | --- | --- | --- | --- | --- | --- |
> > > | **Acc** | 0.0 | 6.2 | 41.7 | 53.3 | 52.5 | 53.7 | 55.4 | 52.9 | 39.6 | 0.8 |
> > > | **Length** |Repetitive generation| 1941.2 | 6232.5 | 8735.6 | 12451.2 | 14364.6 | 15144.8 | 15843.4 | 20241.7 |Repetitive generation|

---

> > > > ### Author Response · Authors · 2025-08-08
> > > >
> > > > Dear Reviewer eb6B,
> > > >
> > > > We hope this message finds you well. As the discussion period is nearing its end with **less than 48 hours remaining**, we wanted to check in to ensure that we have adequately addressed your concerns. Your constructive feedback and suggestions are very important to us, and we would greatly appreciate any further thoughts you might have.
> > > >
> > > > Thank you again for your time and effort in reviewing our paper and rebuttal.

---

> > > > ### Comment · Reviewer_eb6B · 2025-08-08
> > > >
> > > > My concerns are adequately addressed. I am willing to increase my score to 4.

---

> > > > > ### Author Response · Authors · 2025-08-08
> > > > >
> > > > > Thank you for taking the time to review our rebuttal and follow-up comment! We sincerely appreciate your thoughtful feedback throughout the process, and we will continue to strengthen our work based on your valuable suggestions as well as those from the other reviewers.

---

### Official Review · Reviewer_CQee · 2025-07-03

**Clarity:** 3
**Significance:** 2
**Originality:** 3
**Rating:** 5
**Confidence:** 4

**Summary:**

This paper proposes using steering vector to control the reasoning speed of language models. It constructs "slow" and "fast" reasoning responses by prompting models with specific keywords and then creates representations for reasoning mode switching. Additionally, they introduce a mechanism to dynamically adjust the model's reasoning behavior by estimating question difficulty through token-level logit variation. Experimental results demonstrate that adaptive control via the steering vector can improve performance while reducing the number of tokens used.

**Questions:**

1. How frequently do models switch between reasoning modes? It would be helpful to include some analysis showing how often the model transitions between fast and slow modes during inference.
2. Is the steering vector always applied? Are there cases where the steering coefficient $\alpha$ is zero, preserving the model’s original reasoning behavior?
3. For Qwen-3 model, how is fast reasoning data constructed? Do you try disabling the think mode when building fast reasoning data?

**Ethical Concerns:**

["NO or VERY MINOR ethics concerns only"]

**Final Justification:**

I have reviewed the authors' response and found that the additional analysis addresses my concerns. Overall, I find this work to be an interesting application of steering vectors for controlling model reasoning behavior. I have increased my score.

**Limitations:**

Yes.

**Quality:**

3

**Strengths And Weaknesses:**

Strengths:
1. The paper proposes to use the steering vector to change reasoning behaviors, which effectively controls the fast/slow reasoning behaviors of language models.
2. The research questions addressed are timely and important. Determining when to switch between reasoning modes remains an interesting direction in the community.
3. The writing is well-organized and clear, making the manuscript easy to follow.

Weaknesses:
1. The performance gains appear modest. Across different benchmarks and models, the gains are relatively limited, raising concerns about the practical impact of the proposed method.
2. The method for constructing fast reasoning data is somewhat heuristic. Since keyword-triggered behavior can vary across models, this approach may introduce inconsistencies and limit generalizability.

---

> ### Author Rebuttal · Authors · 2025-07-31
>
> We are glad that you find our work timely and important. We hope our responses below could address your concerns.
> ## W1
> Q: The performance gains appear modest, raising concerns about the practical impact of the proposed method.
>
> **A**: We acknowledge that, as an inference-time method, our performance gains may appear modest compared to optimization-based RL approaches [1,2]. However, we believe the **insights** offered by our work provide meaningful contributions to understanding and improving LLM reasoning. Specifically:
> * We **uncover latent fast-thinking capabilities** in LRMs via prompt-based induction and representation-level control.
> * From an interpretability perspective, our difficulty score and case studies offer a novel lens for **tracking and analyzing** reasoning processes.
> * We highlight a key limitation of existing representation editing techniques[3,4,5], which treat representation control as a static, single-axis intervention problem—typically controlling high-level behaviors (e.g., honesty, helpfulness) globally throughout generations. In contrast, we introduce **temporal adaptivity** as a second control axis, reframing representation intervention as a **dynamic, context-aware process**—a shift that opens new directions for future research.
>
> Our method is highly **extensible** and can be effectively integrated with existing reasoning strategies to achieve **non-trivial improvements**. For example, combining our speed control (Section 3) with parallel search on DeepSeek-R1-Distill-Qwen-32B (using $\alpha=8$) leads to **significant gains** over regular generation, as shown below.
>
> *   AIME24
> |Methods|Pass@8|Pass@16|Pass@32|Pass@64|
> |-|-|-|-|-|
> |Original ($\alpha=0$)|62.1|70.3|76.9|83.3|
> |$\alpha=8$|**69.5**|**75.2**|**79.1**|83.3|
>
>
> *   GPQA Diamond
> |Methods|Pass@8|Pass@16|Pass@32|Pass@64|
> |-|-|-|-|-|
> |Original ($\alpha=0$)|78.3|82.3|85.3|88.4|
> |$\alpha=8$|**82.4**|**87.6**|**91.5**|**94.4**|
> ---
> 1.  O1-Pruner: Length-Harmonizing Fine-Tuning for O1-Like Reasoning Pruning
> 2.  Efficient RL Training for Reasoning Models via Length-Aware Optimization
> 3.  Representation Engineering: A Top-Down Approach to AI Transparency
> 4.  ThinkEdit: Interpretable Weight Editing to Mitigate Overly Short Thinking in Reasoning Models
> 5.  Unlocking General Long Chain-of-Thought Reasoning Capabilities of Large Language Models via Representation Engineering
>
> ## W2
> Q: Since keyword-triggered behavior can vary across models, this approach may introduce inconsistencies and limit generalizability.
>
> **A**: We first investigate the generalizability of keyword-triggered behavior across a diverse set of models using MATH500. We examine the generalizability of keyword-triggered behavior in non-reasoning models (Qwen2.5-7B-Instruct and Llama-3.1-8B-Instruct). Treating their original outputs as fast-thinking, we prompt extended reasoning using the trigger word "Okay" (inspired by Figure 2). This simple cue fails to elicit longer reasoning, likely due to the lack of specialized CoT training.
>
> |**Model**|**Slow-thinking**(Pass@1 (%) / Output Length)|**Fast-thinking** (Pass@1 (%) / Output Length)|
> |-|-|-|
> |Qwen2.5-7B-Instruct|74.8 / 654.0|75.9 / 655.2|
> |Llama-3.1-8B-Instruct|43.9 / 1188.7|41.5 / 702.2|
>
> For reasoning models, we include Llama-3.1-Nemotron-Nano-8B-v1, DeepSeek-R1-Distill-Llama-8B and GLM-Z1-9B-0414. We treat their original responses as slow-thinking outputs and use "To" (consistent with our main paper) to trigger their fast thinkings. The results shown below validate the **generalizability of keyword-triggered behavior across reasoning models**.
> |**Model**| **Slow-thinking**(Pass@1 (%) / Output Length)|**Fast-thinking** (Pass@1 (%) / Output Length) | **Relative Differences**|
> |-|-|-|-|
> |Llama-3.1-Nemotron-Nano-8B-v1|93.7 / 3893.4|91.6 / 2760.6|-2.2% / -29.1%|
> |DeepSeek-R1-Distill-Llama-8B|88.2 / 3694.1|63.8 / 878.7|-27.7% / -76.2%|
> |GLM-Z1-9B-0414|96.0 / 2847.1|91.7 / 1691.8|-4.5% / -40.6%|
>
> We show that our method is highly **generalizable across models**. Using **only** fast and slow-thinking data from DeepSeek-R1-Distill-Qwen-7B as stimuli for **all models**, we extract control vectors from each model and evaluate on AIME24 and MATH500. The consistent scaling effects suggest that fast and slow thinking are **intrinsic, shared capabilities of LLMs**. This transferability (1) removes the need for model-specific stimuli generation and (2) avoids inconsistencies stemming from keyword-triggered behaviors.
>
> | Models | AIME 24 ( Pass@1 (%) / Output Length )|  |  |  |  |
> | -|-|-|-| - |-|
> | |$\alpha=8$| $\alpha=4$|$\alpha=0$| $\alpha=-2$ | $\alpha=-4$ |
> |**Non-reasoning** | | | | | |
> |Llama3.1-8B-Instruct |**8.3** / 783.8 | 5.4 / 1293.2 | 6.8 / 3501.2 | 6.7 / 3787.6 | **7.1** / 4748.7 |
> |Qwen2.5-7B-Instruct |9.2 / 1574.9 | **10.4** / 1841.0 | 9.6  /  1886.7 | **11.7** / 1959.4 | **10.8** / 3177.7 |
> |**Reasoning**| | | | | |
> | Llama-3.1-Nemotron-Nano-8B-v1 | 47.1 / 8072.1 | 62.1/ 10719.1 | 62.9 / 11044.5 | **65.0** / 11682.1 | **63.3** / 13414.8 |
> | DeepSeek-R1-Distill-Llama-8B | 35.0 / 10589.4 | 42.1 / 11164.8 | 50.0 / 13398.3 | **53.3** / 14709.1 | **51.7** / 15776.1 |
> | GLM-Z1-9B-0414 | 65.0 / 7870.0 | 67.1 / 8346.8 | 67.5 / 8883.45 | **71.3** / 9585.57 | **72.1** / 10150.1 |
>
> | Models | MATH500 ( Pass@1 (%) / Output Length ) |  |  |  |  |
> | --- | --- | --- | --- | --- | --- |
> |  | $\alpha=8$ | $\alpha=4$ | $\alpha=0$ | $\alpha=-2$ | $\alpha=-4$ |
> | **Non-reasoning** |  |  |  |  |  |
> | Qwen2.5-7B-Instruct | 72.9 / 534.6 | 74.0 / 544.2 | 75.9 / 655.2 | **76.2** / 700.3 | 75.4 / 868.0 |
> | Llama3.1-8B-Instruct | **41.6** / 541.8 | **42.9** / 599.4 | 41.5 / 702.2 | **42.8** / 773.9 | 39.5 / 815.2 |
> | **Reasoning**  |  |  |  |  |  |
> | Llama-3.1-Nemotron-Nano-8B-v1 | 79.4 / 2117.6 | 87.7 / 3091.64 | 93.7 / 3893.4 | **94.3** / 4118.9 | **94.3** / 4303.09 |
> | DeepSeek-R1-Distill-Llama-8B | 75.9 / 2650.7 | 85.1 / 2895.3 | 88.2 / 3694.1 | **89.6** / 4241.2 | **90.2** / 4829.1 |
> | GLM-Z1-9B-0414 | 94.3 / 2215.0 | **96.2** / 2439.9 | 96.0 / 2847.1 | **96.5** / 3023.3 | **96.4** / 3265.5 |
>
> ## Q1
> Q: How frequently do models switch between reasoning modes? It would be helpful to include some analysis.
>
> **A**: Thank you for your insightful suggestion. We analyze reasoning mode switch frequency for DeepSeek-R1-Distill-Qwen-7B on math benchmarks, finding averages of **578.6** (AIME24) and **167.2** (MATH500) switches per question. The higher frequency for AIME24 reflects its greater complexity, requiring more slow-thinking engagement.
>
> Next, to better understand the model's reasoning dynamics, we examine **how switching frequency varies throughout the thought process**. We divide the responses into three temporal segments: start (first 25%), middle (25%-75%), and end (last 25%). We then calculate the average token intervals between these switches (with lower values indicating more frequent switches). The results are presented below:
> | **Dataset** | | | Avg. token interval between switches|
> |-|-|-|-|
> ||Start|Mid|End|
> |AIME24|17.8|20.9|54.9|
> |MATH500|14.5|22.5|41.7|
>
> We found that (1) the model most frequently enters slow-thinking mode at the beginning, likely due to initial problem analysis requiring deeper processing; (2) the middle segment also shows high switching frequency, reflecting active reasoning during problem-solving; and (3) switching drops in the final segment, as the model converges on a solution and generates fewer new thoughts.
>
> Finally, we analyze **the relationship between internal thinking modes and model outputs**. First, we identify the top-100 tokens that most frequently trigger slow-thinking transitions:
> | **Type** | Most Frequent Words |
> |-|-|
> | Calculation | sqrt, denominator, ≈, triangle, product, ... |
> | Analysis | Problem, would, seems, because, find, ... |
> | Reflection | Wait, Alternatively, no, maybe ... |
>
> Our results demonstrate that models tend to switch to slow-thinking for (1) mathematical computations, (2) logical deductions, and (3) triggering certain reflection behaviors.
>
> To assess **the alignment between internal mode switches and external outputs**, we measured how often the token "Wait", commonly used as a marker of uncertainty [1,2], coincides with actual transitions to slow-thinking mode. Surprisingly, on AIME24, the model outputs "Wait" 55.3 times per question on average, but only **1 in 12.2** instances aligns with a true mode switch. On MATH500, the ratio is **1 in 15.0**.
>
> These results suggest that (1) the overuse of "Wait" reflects the **overthinking behavior**[1,2], and (2) analyses relying on output tokens to infer reasoning states [1,3] may be **misleading**, given the weak correlation with internal cognitive transitions.
>
> We hope these findings will provide additional insights for interpreting LLMs' reasoning behaviors and inspire future research in this area.
>
> ---
> 1. Do NOT Think That Much for 2+3=? On the Overthinking of o1-Like LLMs
> 2. Efficient Reasoning Through Suppression of Self-Affirmation Reflections in Large Reasoning Models
> 3. Thoughts Are All Over the Place: On the Underthinking of o1-Like LLMs
>
> ## Q2
> Q: Is the steering vector always applied? Are there cases where the steering coefficient is zero, preserving the model’s original reasoning behavior?
>
> **A**: The steering coefficient can be zero. Our sliding-window control operates in the following logic:
> ```
> if current_difficulty_score > current_detection_threshold:
>     current_steering_intensity = minimum_steering_intensity
> else:
>     current_steering_intensity += acceleration_step_size
> ```
> Since the minimum steering intensity is negative, the intensity will gradually increase (passing through zero) when no high-difficulty signals are detected.
>
> ## Q3
> Q: For Qwen-3 model, how is fast reasoning data constructed?
>
> **A**: We followed the same protocol described in Section 2 to generate fast-thinking responses (i.e., prompting the model to begin its reasoning with "To"), consistent with all other models in our main experiments.

---

> > ### Comment · Reviewer_CQee · 2025-08-05
> >
> > I have reviewed the authors' response and found that the additional analysis addresses my concerns. I encourage the authors to incorporate these explanations into the final revision. Overall, I find this work to be a good application of steering vectors for controlling model reasoning behavior. I have raised my score accordingly.

---

> > > ### Author Response · Authors · 2025-08-05
> > >
> > > We sincerely appreciate the reviewer's positive evaluation and constructive feedback. We are also delighted to know that our additional analyses have adequately addressed your concerns. These explanations, along with the accompanying experimental results, will be included in our final version of the work.

---

### Official Review · Reviewer_kUJX · 2025-07-03

**Clarity:** 4
**Significance:** 3
**Originality:** 3
**Rating:** 5
**Confidence:** 4

**Summary:**

This paper characterizes the distinction between slow / fast thinking in LLMs, extracts the steering vector capable of controlling thinking speed, and proposes a dynamic decoding method involving difficulty estimation to improve reasoning accuracy while reducing token usage.

**Questions:**

The main question is to address weakness 1:
1. Can you outline the main challenges you faced during this work and what innovations, if any, are employed to overcome these challenges?

However, I will most likely not lower my score as long as the following questions are addressed.

2. Can you discuss the claim "plug-and-play" and the time cost of obtaining the reasoning strategy for a pre-trained model?
3. In Step 3 of 3.1 (Line 142), the PCA step of separating hidden state pairs into two halves seem unnecessary. At first glance, the exact same result can be achieved by calculating the first principal component of all $\{ d_i ^{(- \rightarrow +)} \}$'s, at least mathematically. Can you discuss on this, either justify the split or revise this paragraph?
4. Can you further discuss the use of a sliding-window for difficulty evaluation?
5. It would be great if you can discuss the effects of extreme values of steering intensity $\alpha$.

**Ethical Concerns:**

["NO or VERY MINOR ethics concerns only"]

**Final Justification:**

The authors defended their paper on my major concerns, so I maintain my view that this is a solid paper worthy of acceptance.

**Limitations:**

yes

**Quality:**

4

**Strengths And Weaknesses:**

Strengths:
The paper is very well-written, with clear structures, great presentation and thorough justifications. Each step is backed by extensive experimentation. Good job!

Weaknesses:
1. My main reservation is the significance of this work. While the claims are all clear and significant, it seems to me the methods employed are natural and a little lacking in novelty.
2. The method is not exactly one-shot: it requires forward passing a dataset of slow vs. fast thinking CoTs to extract the steering vector specific to the model. Even though this does not require additional training, I'm not sure if it qualifies as plug-and-play.
3. Some minor issues, see Questions.

---

> ### Author Rebuttal · Authors · 2025-07-31
>
> We are glad that you find our work solid and well-written. We hope our responses below will address your concerns.
> ## W1 & Q1
> Q: My main reservation is the significance of this work. Can you outline the main challenges you faced during this work and what innovations are employed to overcome these challenges?
>
> **A**: Thank you for your thoughtful comment. We believe the main challenge—and the key innovation of our method—lies in transforming the _top-down, global, and static_ intervention approaches used in prior works [1,2,3] into a **bottom-up, fine-grained** control mechanism that is both **query-adaptive and reasoning-aware**.
>
> In our early experiments, we found that traditional fixed intervention methods fall short in capturing the dynamic nature of human-like reasoning (i.e., the fluid transitions between System 1 and System 2 thinking). This stems from a key limitation: most interventions operate along a _single functional axis_, globally inducing or suppressing certain behaviors across the entire generation. They largely ignore the **temporal dimension**, which is the fact that a model’s internal reasoning states evolve over time. To address this, we introduce a second axis of intervention: **time**. This reframes representation intervention as a dynamic, context-aware process.
>
> For interpretability, our difficulty score metric provides a **novel framework** for analyzing reasoning dynamics. As a reasoning strategy, we extend representation engineering to online, autoregressive generation with **temporal adaptivity**, which opens new possibilities for context-specific alignment and efficiency optimization.
>
> ---
> 1. Representation Engineering: A Top-Down Approach to AI Transparency
> 2. ThinkEdit: Interpretable Weight Editing to Mitigate Overly Short Thinking in Reasoning Models
> 3. Unlocking General Long Chain-of-Thought Reasoning Capabilities of Large Language Models via Representation Engineering
>
> ## W2 & Q2
> Q: The method is not exactly one-shot. Can you discuss the claim "plug-and-play" and the time cost of obtaining the reasoning strategy for a pre-trained model?
>
> **A**: We use "plug-and-play" to highlight our method's **seamless** integration into LLM frameworks, offering 3 advantages:
> 1. No model structure changes (simple plug-in)
> 2. No model constraints (broad generalizability)
> 3. No parameter adjustments (zero training cost)
>
> Point 1 is self-explanatory. For Point 2, we tested across diverse architectures (reasoning/non-reasoning models with varying training approaches). To compute the control vector, we reused slow and fast CoTs generated by DeepSeek-R1-Distill-Qwen-7B for representation extraction on all tested models. The results below show consistent scaling effects, confirming the compatibility of our method.
>
> |Models | AIME 24 ( Pass@1 (%) / Output Length )|  |  |  |  |
> |-|-|-|-|-|-|
> || $\alpha=8$|$\alpha=4$ | $\alpha=0$ | $\alpha=-2$ | $\alpha=-4$ |
> | **Non-reasoning** |  |  |  |  |  |
> | Llama3.1-8B-Instruct | **8.3** / 783.8 | 5.4 / 1293.2 | 6.8 / 3501.2 | 6.7 / 3787.6 | **7.1** / 4748.7 |
> | Qwen2.5-7B-Instruct | 9.2 / 1574.9 | **10.4** / 1841.0 | 9.6  /  1886.7 | **11.7** / 1959.4 | **10.8** / 3177.7 |
> | **Reasoning** |  |  |  |  |  |
> | Llama-3.1-Nemotron-Nano-8B-v1 | 47.1 / 8072.1 | 62.1/ 10719.1 | 62.9 / 11044.5 | **65.0** / 11682.1 | **63.3** / 13414.8 |
> | DeepSeek-R1-Distill-Llama-8B | 35.0 / 10589.4 | 42.1 / 11164.8 | 50.0 / 13398.3 | **53.3** / 14709.1 | **51.7** / 15776.1 |
> | GLM-Z1-9B-0414 | 65.0 / 7870.0 | 67.1 / 8346.8 | 67.5 / 8883.45 | **71.3** / 9585.57 | **72.1** / 10150.1 |
>
> | Models | MATH500 ( Pass@1 (%) / Output Length ) |  |  |  |  |
> |-|-|-|-|-|-|
> |  | $\alpha=8$ | $\alpha=4$ | $\alpha=0$ | $\alpha=-2$ | $\alpha=-4$ |
> | **Non-reasoning** |  |  |  |  |  |
> | Qwen2.5-7B-Instruct |72.9 / 534.6|74.0 / 544.2|75.9 / 655.2|**76.2** / 700.3|75.4 / 868.0|
> | Llama3.1-8B-Instruct |**41.6** / 541.8| **42.9** / 599.4 | 41.5 / 702.2|**42.8** / 773.9|39.5 / 815.2|
> | **Reasoning**  |  |  |  |  |  |
> | Llama-3.1-Nemotron-Nano-8B-v1|79.4 / 2117.6 | 87.7 / 3091.64 | 93.7 / 3893.4 | **94.3** / 4118.9 | **94.3** / 4303.09 |
> | DeepSeek-R1-Distill-Llama-8B|75.9 / 2650.7 | 85.1 / 2895.3 | 88.2 / 3694.1 | **89.6** / 4241.2 | **90.2** / 4829.1 |
> | GLM-Z1-9B-0414 | 94.3 / 2215.0| **96.2** / 2439.9 | 96.0 / 2847.1 | **96.5** / 3023.3 | **96.4** / 3265.5 |
>
> Regarding Point 3 and **time costs**, our extraction process consists of three main steps:
> 1. Slow and fast CoTs generation (2 hours);
> 2. Hidden states computation (**10 minutes**);
> 3. PCA computation (**few seconds**).
>
> Timings were measured on an NVIDIA L20 46GB×8 server using DeepSeek-R1-Distill-Qwen-7B and MATH training set (7.5k questions) as stimuli. With reusable stimulus pairs, adaptation to new models takes **minutes** - far faster than training-based methods [1,2].
>
> We'll open-source pre-computed control vectors for popular models and slow and fast thinking CoTs for representation extraction to enhance plug-and-play usability. We will reconsider the term "plug-and-play" in the final version of our paper.
>
> ---
> 1. CoT-Valve: Length-Compressible Chain-of-Thought Tuning
> 2. O1-Pruner: Length-Harmonizing Fine-Tuning for O1-Like Reasoning Pruning
>
> ## Q3
> Q: In Step 3 of 3.1 (Line 142), the PCA step of separating hidden state pairs into two halves seem unnecessary. The exact same result can be achieved by calculating the first principal component of all $d_i^{(-\rightarrow+)}$'s.
>
> **A**:  PCA computes principal eigenvectors from the covariance matrix, which captures data dispersion around the mean. Splitting the vector set $\{d_i^{(-\rightarrow+)}\}$ and reversing half **shifts the centroid and changes the dataset’s geometry**, likely resulting in a different covariance matrix and principal components compared to the original $\{d_i^{(-\rightarrow+)}\}$.
>
> As a simple example, consider four data points: $\{1, 1, -1, -1\}$, whose principal direction lies along the x-axis. Flipping the signs of the last two yields $\{1, 1, 1, 1\}$, collapsing the variance to zero and eliminating any principal direction.
>
> Returning to our experiment in Section 3, our goal is to extract a direction that captures the transition from slow to fast thinking. While this direction may already align with a principal component of $\{d_i^{(-\rightarrow+)}\}$, by splitting the dataset and reversing half of it, we **amplify the variance** along the slow->fast axis. This introduces a separating hyperplane, making the target direction easier to isolate and more robust.
>
> To test this, we conduct a controlled experiment using the principal component of $\{d_i^{(-\rightarrow+)}\}$ as a baseline vector. We compare its performance with our method on DeepSeek-R1-Distill-Qwen-7B using AIME24. Results are shown below, where \(\alpha\) is the intervention intensity (as in Figure 5), and each cell reports **Pass@1** (%) ↑ / **Output Length** ↓.
>
> |  | $\alpha=16$ | $\alpha=8$ | $\alpha=4$ | $\alpha=0$ | $\alpha=-4$ |
> | --- | --- | --- | --- | --- | --- |
> | **Baseline** | 46.7 / 10447.0 | 52.5 / 11493.8 | 54.6 / 12367.3 | 52.5 / 12451.2 | 52.5 / 14747.4 |
> | **Our Paper** | 41.7 / **6232.5** | **53.3** / **8735.6** | 50.4 / **10942.7** | 52.5 / 12451.2  | **55.4/** 15144.8 |
>
> As shown above, our control vector consistently achieves more effective length control and boosts accuracy when thoughts are extended, validating the effectiveness of our extraction protocol.
>
> ## Q4
> Q: Can you further discuss the use of a sliding-window for difficulty evaluation?
>
> **A**: The sliding-window mechanism detects shifts from straightforward to complex reasoning via sudden increases in the difficulty score (Eq. 4). For such outlier detection, we use a dynamic threshold, defined as the mean plus standard deviation within the window. This method offers two key advantages:
>
> * **Generalizability**: Different models and tasks produce varying difficulty score scales. The adaptive threshold removes the need for manual tuning and works robustly across benchmarks.
>
> * **Accuracy–Efficiency Trade-off**: As shown in Figure 6 and Table 1, hard reasoning often begins with a spike in difficulty followed by sustained high scores. Our method captures this onset, which guarantees necessary pauses to encourage reflection, and avoids over-intervention afterward. Table 2 and Figures B1–B3 validate this balance.
>
> We provide comprehensive ablation studies in Appendix B to validates the effectiveness of our method. Additionally, we compare our sliding-window approach with a **fixed-threshold baseline**, where the threshold is set to the average difficulty score computed on each test benchmark. The results (shown below) show that our sliding-window method consistently yields a better accuracy–efficiency trade-off.
>
> | **Methods** | AIME24 (Pass@1 (%) / Output Length) | MATH500 (Pass@1 (%) / Output Length) |
> | --- | --- | --- |
> | Original Inference | 52.5 / 12451.2 | 92.9 / 3403.9 |
> | Sliding-window control | **53.8** / **10850.9** | **93.7** / **3122.8** |
> | Fix-threshold based control | 52.1 / 13207.4 | **94.1** / 3811.72 |
>
> ## Q5
> Q: It would be great if you can discuss the effects of extreme values of steering intensity $\alpha$.
>
> **A**: To show the effects of extreme values of steering intensity $\alpha$, we extend our experiments in Figure 5 by scaling up the absolute value of $\alpha$ when controlling the thinking speed of DeepSeek-R1-Distill-Qwen-7B on AIME24. The results are shown below:
> |  | $\alpha=64$ | 32 | 16 | 8 | 0 | -2 | -4 | -6 | -16 | -32 |
> |-|-|-|-|-|-|-|-|-|-|-|
> | **Acc** |0.0|6.2|41.7|53.3|52.5| 53.7 | 55.4 | 52.9 | 39.6 | 0.8 |
> | **Length** |Repetitive generation|1941.2|6232.5|8735.6|12451.2|14364.6|15144.8|15843.4|20241.7|Repetitive generation|
>
> As shown in the table, increasing $\alpha>0$ leads to progressively shorter responses. And when $\alpha$ becomes too large, we observe repetitive generation behavior. Similarly, for $\alpha<0$, increasing $|\alpha|$ produces longer responses, eventually causing repetition.

---

> > ### Comment · Reviewer_kUJX · 2025-08-05
> > **Response to Rebuttal**
> >
> > I thank the authors for their detailed and convincing rebuttals, addressing all of my questions and concerns thoroughly. I have no more questions and would keep my score.
> >
> > To those concerned, I would like to note that my initial comment on PCA was wrong and unwarranted, and I apologize to the authors for the mistake on my part.

---

> > > ### Author Response · Authors · 2025-08-05
> > >
> > > We sincerely thank the reviewer for the thoughtful engagement and kind follow-up. We greatly appreciate your acknowledgment and are glad that our responses were helpful in addressing your concerns. Your comments and suggestions have contributed meaningfully to improving the clarity and rigor of our work.

---

### Decision · Program_Chairs · 2025-09-17

**Decision:**

Accept (spotlight)

**Comment:**

This paper proposes to control the reasoning speed of language models using steering vectors derived from prompting. It also introduces a method to automatically adjust the model's reasoning speed by question difficulty estimate via token-level logit variation.

The idea is novel and interesting, and the experiments are comprehensive. The reviewers raised some questions about the generality and theoretic justifications of the proposed methods. The authors did a good job in clearing those concerns.